# Multi-Agent Reinforcement Learning is A Sequence Modeling Problem

**Muning Wen**[1,2]**, Jakub Grudzien Kuba**[3]**, Runji Lin**[4]**,**
**Weinan Zhang**[1]**, Ying Wen**[1]**, Jun Wang**[2,5]**, Yaodong Yang**[6,7,†]
[1]Shanghai Jiao Tong University, [2]Digital Brain Lab, [3]University of Oxford,
[4]Institute of Automation, Chinese Academy of Science,
[5]University College London, [6]Beijing Institute for General AI,
[7]Institute for AI, Peking University

## Abstract

Large sequence models (SM) such as GPT series and BERT have displayed outstanding performance and generalization capabilities in natural language process, vision and recently reinforcement learning. A natural follow-up question is how to abstract multi-agent decision making also as an sequence modeling problem and benefit from the prosperous development of the SMs. In this paper, we introduce a novel architecture named Multi-Agent Transformer (MAT) that effectively casts cooperative multi-agent reinforcement learning (MARL) into SM problems wherein the objective is to map agents' observation sequences to agents' optimal action sequences. Our goal is to build the bridge between MARL and SMs so that the modeling power of modern sequence models can be unleashed for MARL. Central to our MAT is an encoder-decoder architecture which leverages the multi-agent advantage decomposition theorem to transform the joint policy search problem into a sequential decision making process; this renders only linear time complexity for multi-agent problems and, most importantly, endows MAT with monotonic performance improvement guarantee. Unlike prior arts such as Decision Transformer fit only pre-collected offline data, MAT is trained by online trial and error from the environment in an on-policy fashion. To validate MAT, we conduct extensive experiments on StarCraftII, Multi-Agent MuJoCo, Dexterous Hands Manipulation, and Google Research Football benchmarks. Results demonstrate that MAT achieves superior performance and data efficiency compared to strong baselines including MAPPO and HAPPO. Furthermore, we demonstrate that MAT is an excellent few-short learner on unseen tasks regardless of changes in the number of agents. See our project page at https://sites.google.com/view/multi-agent-transformer[(1)].

## 1 Introduction

Multi-agent reinforcement learning (MARL) [44, 8] is a challenging problem for its difficulty which arises not only from identifying each individual agent's policy improvement direction, but also from combining agents' policy updates jointly which should be beneficial for the whole team. Recently, such difficulty in multi-agent learning has been eased owing to the introduction of *centralized training for decentralized execution* (CTDE) [11, 45], which allows agents to access the global information and opponents' actions during the training phase. This framework enables successful developments of methods that directly inherit single-agent algorithms. For examples, COMA replaces the policy-gradient (PG) estimate with a multi-agent PG (MAPG) counterpart [11], MADDPG

---

[(1)][†]Corresponding to <yaodong.yang@pku.edu.cn>. The source code could be accessed directly with this link https://github.com/PKU-MARL/Multi-Agent-Transformer.

extends deterministic policy gradient into multi-agent settings with a centralized critic [20, 34], QMIX leverages deep Qnetworks for decentralized agents and introduces a centralized mixing network for Q-value decomposition [29, 36, 26]. MAPPO endowing all agents with the same set of parameters and then training by trust-region methods [46]. PR2 [42] and GR2 [43] methods conduct recursive reasoning under the CTDE framework. These methods, however, cannot cover the whole complexity of multi-agent interactions; in fact, some of them are shown to fail in the simplest cooperative task [15]. To resolve this issue, the *multi-agent advantage decomposition theorem* was proposed [15, Theorem 1] which captures how different agents contribute to the return and provides an intuition behind the emergence of cooperation through a sequential decision making process scheme. Based on it, HATRPO and HAPPO algorithms [15, 17, 16] were derived which, thanks to the decomposition theorem and sequential update scheme, established new state-of-the-art methods for MARL. However, their limitation is that the agents' policies are unaware of the purpose to develop cooperation and still rely on a carefully handcrafted maximization objective. Ideally, a team of agents should be aware of the jointness of their training by design, thereby following a holistic and effective paradigm—an ideal solution that is yet to be proposed.

In recent years, *sequence models* (SM) have made a substantial progress in *natural language processing* (NLP) [27]. For example, GPT series [3] and BERT models [9], built on autoregressive SMs, have demonstrated remarkable performance on a wide range of downstream tasks and achieved strong performance on few-shot generalization tasks. Although SM are mostly used in language tasks due to its natural fitting with the sequential property of languages, the sequential approaches are not confined to NLP only, but is instead a widely applicable general foundation model [2]. For example, in *computer vision* (CV), one can split an image into sub-images and align them in a sequence as if they were tokens in NLP tasks [9, 10, 12]. Although the idea of solving CV tasks by SM is straightforward, it serves as the foundation to some of the best-performing CV algorithms [38, 41, 39]. Furthermore, sequential methods are starting to spawn powerful multi-modal visual language models such as Flamingo [1], DALL-E [28], and GATO [30] in the recent past.

Coming with effective and expressive network architectures such as *Transformer* [40], sequence modeling techniques have also attracted tremendous attention from the RL community, which results in a series of successful offline RL developments based on the Transformer architecture [5, 14, 30, 23]. These methods show great potentials in tackling some of the most fundamental RL training problems, such as long-horizon credit assignment and reward sparsity [37, 24, 25]. For example, by training autoregressive models on pre-collected offline data in a purely supervised way, Decision Transformer [5] bypasses the need for computing cumulative rewards through dynamic programming, but rather generates future actions conditioning on the desired returns, past states and actions. Despite their remarkable successes, none of these methods have been designed to model the most difficult (also unique to MARL) aspect of multi-agent systems—the agents' interactions. In fact, if we were to simply endow all agents with a Transformer policy and train them independently, their joint performance still could not be guaranteed to improve [15, Proposition 1]. Therefore, while a myriad of powerful SMs are available, MARL—an area that would greatly benefit from SM—has not truly taken advantage of their performance benefit. The key research question to ask is then

**How can we model MARL problems by sequence models ?**

In this paper, we take several steps to provide an affirmative answer to the above research question. Our goal is to enhance MARL studies with powerful sequential modeling techniques. To fulfill that, we start by proposing a novel MARL training paradigm which establishes the connection between cooperative MARL problems and sequence modeling problems. Central to the new paradigm are the *multi-agent advantage decomposition theorem* and *sequential update scheme*, which effectively transform multi-agent joint policy optimization into a sequential policy search process. As a natural outcome of our findings, we introduce *Multi-Agent Transformer* (MAT), an encoder-decoder architecture that implements generic MARL solutions through SM. Unlike Decision Transformer [5], MAT is trained online based on trials and errors in an on-policy fashion; therefore, it does not require collecting demonstrations upfront. Importantly, the implementation of the multi-agent advantage decomposition theorem ensures MAT to enjoy monotonic performance improvement guarantee during training. MAT establishes a new state-of-the-art baseline model for cooperative MARL tasks. We justify such a claim by evaluating MAT on the benchmarks of StarCraftII, Multi-Agent MuJoCo, Dexterous Hands Manipulation, and Google Research Football; results show that MAT achieves superior performance over strong baselines, such as MAPPO [46], HAPPO [15], QMIX [29] and UPDeT [13]. Finally, we show that MAT possesses great potentials in task generalizations, which holds regardless of the agent number in new tasks.

## 2 Preliminaries

In this section, we first introduce the cooperative MARL problem formulation and the multi-agent advantage decomposition theorem, which serves as the cornerstone of our work. We then review existing MARL methods that relate to MAT, and finally familiarize the reader with the Transformer.

### 2.1 Problem Formulation

Cooperative MARL problems are often modeled by *Markov games* $\langle \mathcal{N}, \mathcal{O}, \mathcal{A}, R, P, \gamma \rangle$ [19]. $\mathcal{N} = \{1, \ldots, n\}$ is the set of agents, $\mathcal{O} = \prod_{i=1}^{n} \mathcal{O}^i$ is the product of local observation spaces of the agents, namely the joint observation space, $\mathcal{A} = \prod_{i=1}^{n} \mathcal{A}^i$ is the product of the agents' action spaces, namely the joint action space, $R : \mathcal{O} \times \mathcal{A} \to [-R_{\max}, R_{\max}]$ is the joint reward function, $P : \mathcal{O} \times \mathcal{A} \times \mathcal{O} \to \mathbb{R}$ is the transition probability function, and $\gamma \in [0, 1)$ is the discount factor. At time step $t \in \mathbb{N}$, an agent $i \in \mathcal{N}$ observes an observation $o_t^i \in \mathcal{O}^i$ [(2)] ($o = (o^1, \ldots, o^n)$ is a "joint" observation) and takes an action $a_t^i$ according to its policy $\pi^i$, which is the $i^{\text{th}}$ component of the agents' joint policy $\boldsymbol{\pi}$. At each time step, all agents take actions **simultaneously** based on their observation with no sequential dependencies. The transition kernel $P$ and the joint policy induce the (improper) marginal observation distribution $\rho_{\boldsymbol{\pi}}(\cdot) \triangleq \sum_{t=0}^{\infty} \gamma^t \Pr(\mathbf{o}_t = \boldsymbol{o} | \boldsymbol{\pi})$. At the end of each time step, the whole team receives a joint reward $R(\mathbf{o}_t, \mathbf{a}_t)$ and observe $\mathbf{o}_{t+1}$, whose probability distribution is $P(\cdot | \mathbf{o}_t, \mathbf{a}_t)$. Following this process infinitely long, the agents earn a discounted cumulative return of $R^\gamma \triangleq \sum_{t=0}^{\infty} \gamma^t R(\mathbf{o}_t, \mathbf{a}_t)$.

### 2.2 Multi-Agent Advantage Decomposition Theorem

The agents evaluate the value of actions and observations with $Q_{\boldsymbol{\pi}}(\boldsymbol{o}, \boldsymbol{a})$ and $V_{\boldsymbol{\pi}}(\boldsymbol{o})$, defined as

$$Q_{\boldsymbol{\pi}}(\boldsymbol{o}, \boldsymbol{a}) \triangleq \mathbb{E}_{\mathbf{o}_{1:\infty} \sim P, \mathbf{a}_{1:\infty} \sim \boldsymbol{\pi}} [R^\gamma | \mathbf{o}_0 = \boldsymbol{o}, \mathbf{a}_0 = \boldsymbol{a}], \tag{1}$$
$$V_{\boldsymbol{\pi}}(\boldsymbol{o}) \triangleq \mathbb{E}_{\mathbf{a}_0 \sim \boldsymbol{\pi}, \mathbf{o}_{1:\infty} \sim P, \mathbf{a}_{1:\infty} \sim \boldsymbol{\pi}} [R^\gamma | \mathbf{o}_0 = \boldsymbol{o}].$$

The jointness of the objective causes difficulties associated with the *credit assignment* problem—having received a shared reward, individual agents are unable to deduce their own contribution to the team's success or failure [4]. Indeed, applying traditional RL methods which simply employ the above value functions leads to obstacles in training, such as the growing variance of multi-agent policy gradient (MAPG) estimates [17]. Hence, to tackle these, notions of local value functions [21] and counterfactual baselines [11] have been developed. In this paper, we work with the most general notions of this kind—the *multi-agent observation-value functions* [15]. That is, for arbitrary disjoint, ordered subsets of agents $i_{1:m} = \{i_1, \ldots, i_m\}$ and $j_{1:h} = \{j_1, \ldots, j_h\}$, for $m, h \leq n$, we define the multi-agent observation-value function by

$$Q_{\boldsymbol{\pi}}(\boldsymbol{o}, \boldsymbol{a}^{i_{1:m}}) \triangleq \mathbb{E}[R^\gamma | \mathbf{o}_0^{i_{1:n}} = \boldsymbol{o}, \mathbf{a}_0^{i_{1:m}} = \boldsymbol{a}^{i_{1:m}}], \tag{2}$$

which recovers the original state-action value function in Equation (1) when $m = n$, and the original observation-value function when $m = 0$ (*i.e.*, when the set $i_{1:m}$ is empty). Based on Equation (2), we can further measure the contribution of a chosen subset of agents to the joint return and define the multi-agent advantage function by

$$A_{\boldsymbol{\pi}}^{i_{1:m}}(\boldsymbol{o}, \boldsymbol{a}^{j_{1:h}}, \boldsymbol{a}^{i_{1:m}}) \triangleq Q_{\boldsymbol{\pi}}^{j_{1:h}, i_{1:m}}(\boldsymbol{o}, \boldsymbol{a}^{j_{1:h}}, \boldsymbol{a}^{i_{1:m}}) - Q_{\boldsymbol{\pi}}^{j_{1:h}}(\boldsymbol{o}, \boldsymbol{a}^{j_{1:h}}). \tag{3}$$

The above quantity describes how much better/worse than average the joint action $\boldsymbol{a}$ will be if agents $i_{1:m}$ take the joint action $\boldsymbol{a}^{i_{1:m}}$, once $j_{1:h}$ have taken $\boldsymbol{a}^{j_{1:h}}$. Again, when $h = 0$, the advantage compares the value of $\boldsymbol{a}^{i_{1:m}}$ to the baseline value function of the whole team. This value-functional representation of agents' actions enables studying interactions between them, as well to decompose the joint value function signal, thus helping alleviate the severity of the credit assignment problem [29, 35, 22]. The insights of Equation (3) is accomplished by means of the following theorem.

---

[(2)]For notation convenience, we omit defining agents' observation functions that take the global state as the input and outputs a local observation for each agent, but rather define agents' local observations directly.

**Theorem 1** (Multi-Agent Advantage Decomposition [17]). *Let $i_{1:n}$ be a permutation of agents. Then, for any joint observation $\boldsymbol{o} = \boldsymbol{o} \in \mathcal{O}$ and joint action $\boldsymbol{a} = \boldsymbol{a}^{i_{1:n}} \in \mathcal{A}$, the following equation always holds with no further assumption needed,*

$$A_{\boldsymbol{\pi}}^{i_{1:n}}\big(\boldsymbol{o}, \boldsymbol{a}^{i_{1:n}}\big) = \sum_{m=1}^{n} A_{\boldsymbol{\pi}}^{i_m}\big(\boldsymbol{o}, \boldsymbol{a}^{i_{1:m-1}}, a^{i_m}\big).$$

Importantly, this theorem provides an intuition guiding the choice of incrementally improving actions. Suppose that agent $i_1$ picks an action $a^{i_1}$ with positive advantage, $A_{\boldsymbol{\pi}}^{i_1}(\boldsymbol{o}, a^{i_1}) > 0$. Then, imagine that for all $j = 2, \ldots, n$, agent $i_j$ knows the joint action $\boldsymbol{a}^{i_{1:j-1}}$ of its predecessors. In this case, it can choose an action $a^{i_j}$ for which the advantage $A_{\boldsymbol{\pi}}^{i_j}(\boldsymbol{o}, \boldsymbol{a}^{i_{1:j-1}}, a^{i_j})$ is positive. Altogether, the theorem assures that the joint action $\boldsymbol{a}^{i_{1:n}}$ has positive advantage. Furthermore, notice that the joint action has been chosen in $n$ steps, each of which searched an individual agent's action space. Hence, the complexity of this search is additive, $\sum_{i=1}^{n} |\mathcal{A}^i|$, in the sizes of the action spaces. If we were to perform the search directly in the joint action space, we would browse a set of multiplicative size, $|\mathcal{A}| = \prod_{i=1}^{n} |\mathcal{A}^i|$. Later, we will build upon this insight to design a SM that optimizes joint policies efficiently, agent by agent, without the necessity of considering the joint action space at once.

## 2.3 Existing Methods in MARL

We now briefly summarize two state-of-the-art MARL algorithms. Both of them build upon *Proximal Policy Optimization* (PPO) [33]—a RL method famous for its simplicity and its performance stability.

**MAPPO** [46] is the first, and the most direct, approach for applying PPO in MARL. It equips all agents with one shared set of parameters and use agents' aggregated trajectories for the shared policy's update; at iteration $k + 1$, it optimizes the policy parameter $\theta_{k+1}$ by maximizing the clip objective of

$$\sum_{i=1}^{n} \mathbb{E}_{\mathbf{o} \sim \rho_{\boldsymbol{\pi}_{\theta_k}}, \mathbf{a} \sim \boldsymbol{\pi}_{\theta_k}} \left[ \min \left( \frac{\pi_\theta(\mathbf{a}^i | \mathbf{o})}{\pi_{\theta_k}(\mathbf{a}^i | \mathbf{o})} A_{\boldsymbol{\pi}_{\theta_k}}(\mathbf{o}, \mathbf{a}), \mathrm{clip}\left( \frac{\pi_\theta(\mathbf{a}^i | \mathbf{o})}{\pi_{\theta_k}(\mathbf{a}^i | \mathbf{o})}, 1 \pm \epsilon \right) A_{\boldsymbol{\pi}_{\theta_k}}(\mathbf{o}, \mathbf{a}) \right) \right],$$

where the clip operator clips the input value (if necessary) so that it stays within the interval $[1-\epsilon, 1+\epsilon]$. However, enforcing parameter sharing is equivalent to putting a constraint $\theta^i = \theta^j, \forall i, j \in \mathcal{N}$ on the joint policy space, which can lead to an exponentially-worse sub-optimal outcome [15]. This motivates a more principled development of heterogeneous-agent trust-region methods, e.g., HAPPO.

**HAPPO** [15] is currently one of the SOTA algorithm that fully leverages Theorem (1) to implement multi-agent trust-region learning with monotonic improvement guarantee. During an update, the agents choose a permutation $i_{1:n}$ at random, and then following the order in the permutation, every agent $i_m$ picks $\pi_{\mathrm{new}}^{i_m} = \pi^{i_m}$ that maximizes the objective of

$$\mathbb{E}_{\mathbf{o} \sim \rho_{\boldsymbol{\pi}_{\mathrm{old}}}, \mathbf{a}^{i_{1:m-1}} \sim \boldsymbol{\pi}_{\mathrm{new}}^{i_{1:m-1}}, \mathbf{a}^{i_m} \sim \pi_{\mathrm{old}}^{i_m}} \left[ \min \big( \mathrm{r}(\pi^{i_m}) A_{\boldsymbol{\pi}_{\mathrm{old}}}^{i_{1:m}}(\boldsymbol{o}, \mathbf{a}^{i_{1:m}}), \mathrm{clip}(\mathrm{r}(\pi^{i_m}), 1 \pm \epsilon) A_{\boldsymbol{\pi}_{\mathrm{old}}}^{i_{1:m}}(\mathbf{o}, \mathbf{a}^{i_{1:m}}) \big) \right],$$

where $\mathrm{r}(\pi^{i_m}) = \pi^{i_m}(\mathbf{a}^{i_m} | \mathbf{o}) / \pi_{\mathrm{old}}^{i_m}(\mathbf{a}^{i_m} | \mathbf{o})$. Note that the expectation is taken over the newly-updated previous agents' policies, *i.e,* $\boldsymbol{\pi}_{\mathrm{new}}^{i_{1:m-1}}$; this reflects an intuition that, under Theorem (1), the agent $i_m$ reacts to its preceding agents $i_{1:m-1}$. However, one drawback of HAPPO is that agent's policies has to follow the sequential update scheme in the permutation, thus it cannot be run in parallel.

## 2.4 The Transformer Model

Transformer [40] was originally designed for machine translation tasks (e.g., input English, output French). It maintains an encoder-decoder structure, where the encoder maps an input sequence of tokens to latent representations and then the decoder generates a sequence of desired outputs in an auto-regressive manner wherein at each step of inference, the Transformer takes all previously generated tokens as the input. One of the most essential component in Transformer is the scaled dot-product attention, which captures the interrelationship of input sequences. The attention function is written as $\mathrm{Attention}(\mathbf{Q}, \mathbf{K}, \mathbf{V}) = \mathrm{softmax}\big(\frac{\mathbf{Q}\mathbf{K}^T}{\sqrt{d_k}}\big)\mathbf{V}$, where the $\mathbf{Q}, \mathbf{K}, \mathbf{V}$ corresponds to the vector of queries, keys and values, which can be learned during training, and the $d_k$ represent the dimension of $\mathbf{Q}$ and $\mathbf{K}$. Self-attentions refer to cases when $\mathbf{Q}, \mathbf{K}, \mathbf{V}$ share the same set of parameters.

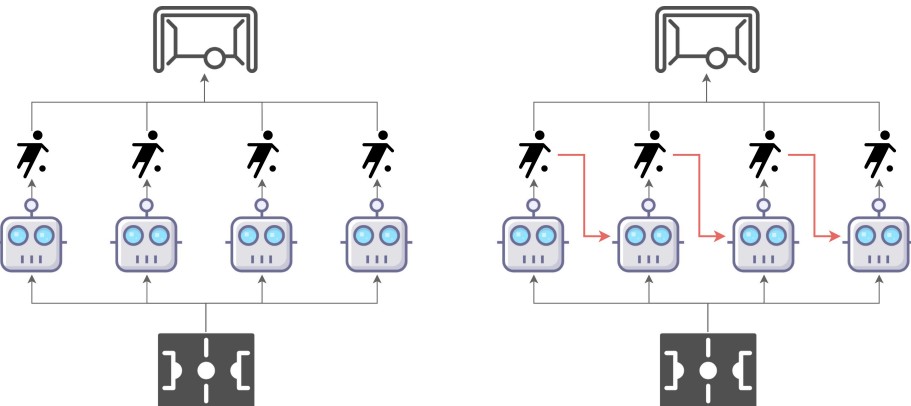

Figure 1: Conventional multi-agent learning paradigm (left) wherein all agents take actions simultaneously *vs.* the multi-agent sequential decision paradigm (right) where agents take actions by following a sequential order, each agent accounts for decisions from preceding agents as red arrows suggest.

Inspired by the attention mechanism, **UPDeT** [13] handles various observation sizes by decoupling each agent's observations into a sequence of observation-entities, matching them with different action-groups, and modeling the relationship between the matched observation-entities with a Transformer-based function for better representation learning in MARL problems. Apart from this, based on the sequential property described in the Theorem (1) and the principle behind HAPPO [15], it is intuitive to think about another Transformer-based implementation for multi-agent trust-region learning. **By treating a team of agents as a sequence**, the Transformer architecture allows us to model teams of agents with variable numbers and types, while avoiding drawbacks of MAPPO/HAPPO. We will describe in more details how a cooperative MARL problem can be solved by a sequence model.

## 3 The Surprising Connection Between MARL and Sequence Models

To establish the connection between MARL and sequence models, Theorem (1) provides a new angle of understanding the MARL problem from a SM perspective. If each agent knows its predecessors' actions with an arbitrary decision order, the sum of agents' local advantages $A_{\boldsymbol{\pi}}^{i_j}(\boldsymbol{o}, \boldsymbol{a}^{i_{1:m-1}}, a^{i_m})$ will be exactly equal to the joint advantages $A_{\boldsymbol{\pi}}^{i_{1:n}}(\boldsymbol{o}, \boldsymbol{a}^{i_{1:n}})$. This orderly decision setting across agents simplifies the update of their joint policy, where maximizing each agent's own local advantage is equivalent to maximizing the joint advantage. As such, agents do not need to worry about interference from other agents anymore during the policy update; the local advantage functions have already captured the relationship between agents. This property revealed by Theorem (1) inspires us to propose a multi-agent sequential decision paradigm for MARL problems as show in Figure (1), where we assign agents with an arbitrary decision order (one permutation for each iteration); each agent can access its predecessors' behaviors, based on which it then takes the optimal decision. This sequential paradigm motivates us to leverage a sequential model, e.g., Transformer, to explicitly capture the sequential relationship between agents described in Theorem (1).

Underpinned by Theorem (1), sequence modeling reduces the complexity growth of MARL problems with the number of agents from multiplicative to additive, thus rendering linear complexity. With the help of the Transformer architecture, we can model policies of heterogeneous agents with an unified network but treat each agent discriminatively with different position, and thus ensuring high sample efficiency while avoiding the exponentially-worse outcome that MAPPO is facing. Besides, in order to guarantee the monotonic improvement of joint policies, HAPPO has to update each policy one-by-one during training, by leveraging previous update results of $\pi^{i_1}, ..., \pi^{i_{m-1}}$ to improve $\pi^{i_m}$, which becomes critical in computational efficiency at large size of agents. By contrast, the attention mechanism of Transformer architectures allows for batching the ground truth actions $a_t^{i_0}, ..., a_t^{i_{n-1}}$ in the buffer to predict $a_t^{i_1}, ..., a_t^{i_n}$ and update policies simultaneously, which significantly improves the training speed and makes it feasible for large size of agents. Furthermore, in cases that the number and the type of agents are different, SM can incorporates them into an unified solution through its capability on modeling sequences with flexible sequence length, rather than treat different agent

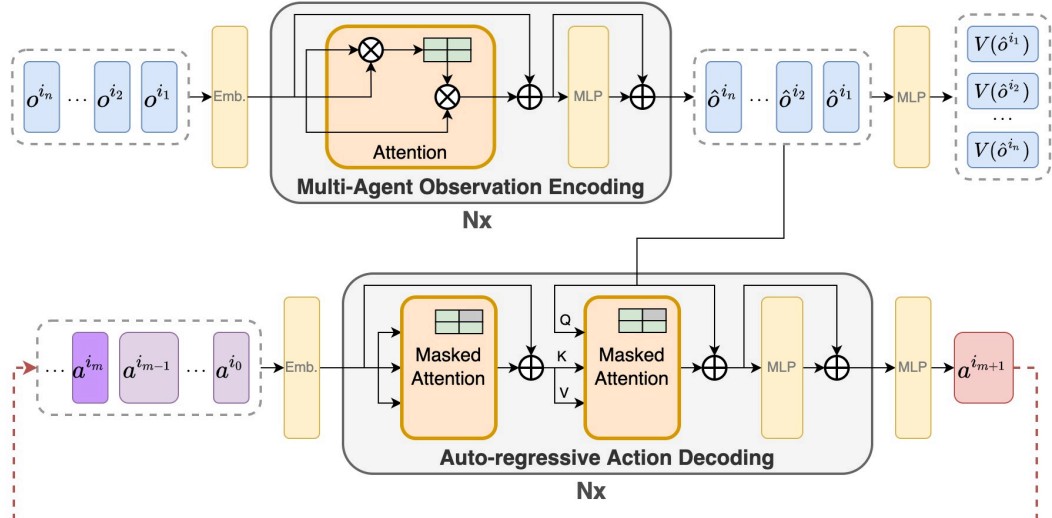

Figure 2: The encoder-decoder architecture of MAT. At each time step, the encoder takes in a sequence of agents' observations and encodes them into a sequence of latent representations, which is then passed into the decoder. The decoder generate each agent's optimal action in a sequential and auto-regressive manner. The masked attention blocks ensures agents can only access its preceding agents' actions during training. We list the full pseudocode of MAT in Appendix A and a video that shows the dynamic data flow of MAT in https://sites.google.com/view/multi-agent-transformer.

numbers as different tasks. To realize the above idea, we introduce a practical architecture named Multi-Agent Transformer in the next section.

## 4 The Multi-Agent Transformer

To implement the sequence modeling paradigm for MARL, our solution is *Multi-Agent Transformer* (MAT). The idea of applying the Transformer architecture comes from the fact that the mapping between the input of agents' observation sequence $(o^{i_1}, \ldots, o^{i_n})$ and the output of agents' action sequence $(a^{i_1}, \ldots, a^{i_n})$ are sequence modeling tasks similar to machine translations. As eluded by Theorem (1), the action $a^{i_m}$ depends on all previous agents' decisions $a^{i_{1:m-1}}$. Hence, our MAT in Figure (2) consists of an *encoder*, which learns representations of the joint observations, and a *decoder* which outputs actions for each individual agent in an auto-regressive manner.

**The encoder**, whose parameters we denote by $\phi$, takes a sequence of observations $(o^{i_1}, \ldots, o^{i_n})$ in arbitrary order and passes them through several computational blocks. Each such block consists of a *self-attention* mechanism and a *multi-layer perceptron* (MLP), as well as *residual connections* to prevent gradient vanishing and network degradation with the increase of depth. We denote the output encoding of the observations as $(\hat{o}^{i_1}, \ldots, \hat{o}^{i_n})$, which encodes not only the information of agents $(i_1, \ldots, i_n)$ but also the high-level interrelationships that represent agents' interactions. In order to learn expressive representations, in the training phase, we make the encoder to approximate the value functions, whose objective is to minimize the empirical Bellman error by

$$L_{\text{Encoder}}(\phi) = \frac{1}{Tn} \sum_{m=1}^{n} \sum_{t=0}^{T-1} \left[ R(\mathbf{o}_t, \mathbf{a}_t) + \gamma V_{\bar{\phi}}(\hat{\mathbf{o}}_{t+1}^{i_m}) - V_\phi(\hat{\mathbf{o}}_t^{i_m}) \right]^2, \qquad (4)$$

where $\bar{\phi}$ is the target network's parameter, which is non-differentiable and updated every few epochs.

**The decoder**, whose parameters we denote by $\theta$, passes the embedded joint action $\mathbf{a}^{i_{0:m-1}}, m = \{1, \ldots n\}$ (where $a^{i_0}$ is an arbitrary symbol indicating the start of decoding) to a sequence of decoding blocks. Crucially, every decoding block comes with a *masked self-attention* mechanism, where the masking makes sure that, for every $i_j$, attention is computed only between the $i_r^{\text{th}}$ and the $i_j^{\text{th}}$ action heads wherein $r < j$ so that the sequential update scheme can be maintained. This is then followed

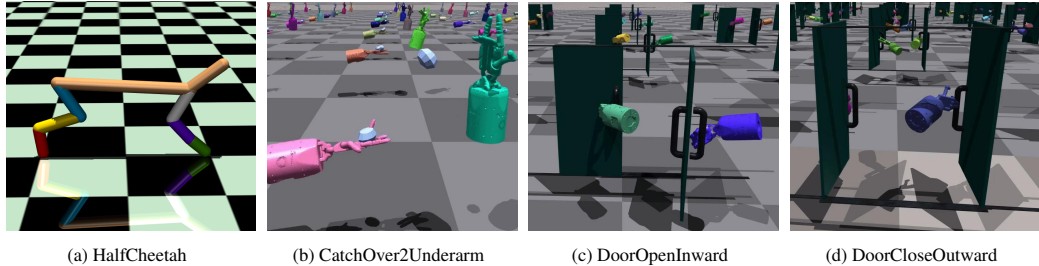

| (a) HalfCheetah | (b) CatchOver2Underarm | (c) DoorOpenInward | (d) DoorCloseOutward |

Figure 3: Demonstrations of the Bi-DexHands and the HalfCheetah environments.

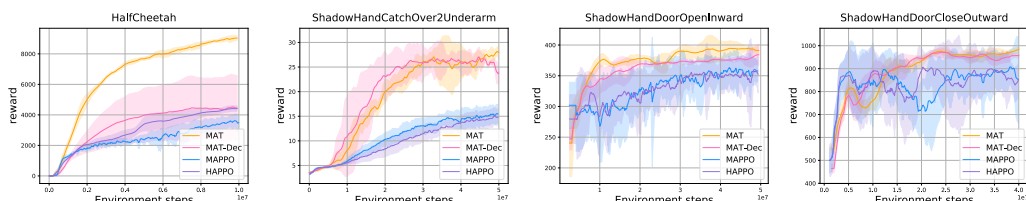

Figure 4: Performance comparisons on the Multi-Agent MuJoCo and the Bi-DexHands benchmarks.

by a second masked attention function, which computes the attention between the action heads and observation representations. Finally, the block finishes with an MLP and skipping connections. The output to the last decoder block is a sequence of representations of the joint actions, $\{\hat{\boldsymbol{a}}^{i_0:i-1}\}_{i=1}^m$. This is fed to an MLP that outputs the probability distribution of $i_m$'s action, namely, the policy $\pi_\theta^{i_m}(\mathrm{a}^{i_m}|\hat{\mathbf{o}}^{i_{1:n}}, \mathrm{a}^{i_{1:m-1}})$. To train the decoder, we minimize the following clipping PPO objective of

$$L_{\text{Decoder}}(\theta) = -\frac{1}{Tn}\sum_{m=1}^{n}\sum_{t=0}^{T-1}\min\left(\mathrm{r}_t^{i_m}(\theta)\hat{A}_t, \text{clip}(\mathrm{r}_t^{i_m}(\theta), 1\pm\epsilon)\hat{A}_t\right), \quad (5)$$

$$\mathrm{r}_t^{i_m}(\theta) = \frac{\pi_\theta^{i_m}(\mathbf{a}_t^{i_m}|\hat{\mathbf{o}}_t^{i_{1:n}}, \hat{\mathbf{a}}_t^{i_{1:m-1}})}{\pi_{\theta_{\text{old}}}^{i_m}(\mathbf{a}_t^{i_m}|\hat{\mathbf{o}}_t^{i_{1:n}}, \hat{\mathbf{a}}_t^{i_{1:m-1}})},$$

where $\hat{A}_t$ is an estimate of the joint advantage function. One can apply *generalized advantage estimation* (GAE) [32] with $\hat{V}_t = \frac{1}{n}\sum_{m=1}^n V(\hat{o}_t^{i_m})$ as a robust estimator for the joint value function. Notably, the action generation process is different between the inference and the training stage. In the inference stage, each action is generated auto-regressively, in the sense that $\mathrm{a}^{i_m}$ will be inserted back into the decoder again to generate $\mathrm{a}^{i_{m+1}}$ (starting with $\mathrm{a}^{i_0}$ and ending with $\mathrm{a}^{i_{n-1}}$). While during the training stage, the output of all actions, $\mathbf{a}^{i_{1:n}}$ can be computed in parallel simply because $\mathbf{a}^{i_{1:n-1}}$ have already been collected and stored in the replay buffer.

**The attention** mechanism, which lies in the heart of MAT, encodes observations and actions with a weight matrix calculated by multiplying the embedded queries, $(q^{i_1}, \ldots, q^{i_n})$, and keys, $(k^{i_1}, \ldots, k^{i_n})$, where each of the weight $w(q^{i_r}, k^{i_j}) = \langle q^{i_r}, k^{i_j}\rangle$. The embedded values $(v^{i_1}, \ldots, v^{i_n})$ are multiplied with the weight matrix to output representations. While the unmasked attention in the encoder uses a full weight matrix to extract the interrelationship between agents, i.e., $\hat{\mathbf{o}}^{i_{1:n}}$, the masked attentions in the decoder capture $\mathbf{a}^{i_{1:m}}$ with triangular matrices where $w(q^{i_r}, k^{i_j}) = 0$ for $r < j$ (see an visual illustration in Appendix A). With the properly masked attention mechanism, the decoder can safely output the policy $\pi_\theta^{i_{m+1}}(\mathbf{a}^{i_{m+1}}|\hat{\mathbf{o}}^{i_{1:n}}, \mathbf{a}^{i_{1:m}})$, which finishes the implementation of Theorem (1).

**The monotonic improvement guarantee.** An MAT agent $i_m$ optimizes a trust-region objective that is conditioned on new decisions of agents $i_{1:m-1}$ by means of conditioning its policy ratio on them (see Equation (5)). As such, it increases the joint return monotonically like if it followed the sequential update scheme of HAPPO [15, Theorem 2]. However, as oppose to that method, the MAT model does not require $i_m$ to wait until its predecessors make their updates, nor it uses their updated action distribution for importance sampling calculations. In fact, as actions of all agents are outputs of MAT, their clipping objectives can be computed in parallel (during training), thus dominating

Table 1: Performance evaluations of win rate and standard deviation on the SMAC benchmark, where UPDeT's official codebase supports several Marine-based tasks only.

| Task | Difficulty | MAT | MAT-Dec | MAPPO | HAPPO | QMIX | UPDeT | Steps |
|---|---|---|---|---|---|---|---|---|
| 3m | Easy | **100.0**$_{(1.8)}$ | **100.0**$_{(1.1)}$ | **100.0**$_{(0.4)}$ | **100.0**$_{(1.2)}$ | 96.9$_{1.3}$ | **100.0**$_{(5.2)}$ | 5e5 |
| 8m | Easy | **100.0**$_{(1.1)}$ | 97.5$_{(2.5)}$ | 96.8$_{(2.9)}$ | 97.5$_{(1.1)}$ | 97.7$_{1.9}$ | 96.3$_{(9.7)}$ | 1e6 |
| 1c3s5z | Easy | **100.0**$_{(2.4)}$ | **100.0**$_{(0.4)}$ | **100.0**$_{(2.2)}$ | 97.5$_{(1.8)}$ | 96.9$_{(1.5)}$ | / | 2e6 |
| MMM | Easy | **100.0**$_{(2.2)}$ | 98.1$_{(2.1)}$ | 95.6$_{(4.5)}$ | 81.2$_{(22.9)}$ | 91.2$_{(3.2)}$ | / | 2e6 |
| 2c vs 64zg | Hard | **100.0**$_{(1.3)}$ | 95.9$_{(2.3)}$ | **100.0**$_{(2.7)}$ | 90.0$_{(4.8)}$ | 90.3$_{(4.0)}$ | / | 5e6 |
| 3s vs 5z | Hard | **100.0**$_{(1.7)}$ | **100.0**$_{(1.3)}$ | **100.0**$_{(2.5)}$ | 91.9$_{(5.3)}$ | 92.3$_{(4.4)}$ | / | 5e6 |
| 3s5z | Hard | **100.0**$_{(1.9)}$ | **100.0**$_{(3.3)}$ | 72.5$_{(26.5)}$ | 90.0$_{(3.5)}$ | 84.3$_{(5.4)}$ | / | 3e6 |
| 5m vs 6m | Hard | **90.6**$_{(4.4)}$ | 83.1$_{(4.6)}$ | 88.2$_{(6.2)}$ | 73.8$_{(4.4)}$ | 75.8$_{(3.7)}$ | **90.6**$_{(6.1)}$ | 1e7 |
| 8m vs 9m | Hard | **100.0**$_{(3.1)}$ | 95.0$_{(4.6)}$ | 93.8$_{(3.5)}$ | 86.2$_{(4.4)}$ | 92.6$_{(4.0)}$ | / | 5e6 |
| 10m vs 11m | Hard | **100.0**$_{(1.4)}$ | **100.0**$_{(2.0)}$ | 96.3$_{(5.8)}$ | 77.5$_{(9.7)}$ | 95.8$_{(6.1)}$ | / | 5e6 |
| 25m | Hard | **100.0**$_{(1.3)}$ | 86.9$_{(5.6)}$ | **100.0**$_{(2.7)}$ | 70.1$_{(8.1)}$ | 90.2$_{(9.8)}$ | 2.8$_{(3.1)}$ | 2e6 |
| 27m vs 30m | Hard+ | **100.0**$_{(0.7)}$ | 95.3$_{(2.2)}$ | 93.1$_{(3.2)}$ | 5.6$_{(2.8)}$ | 39.2$_{(8.8)}$ | / | 1e7 |
| MMM2 | Hard+ | **93.8**$_{(2.6)}$ | 91.2$_{(5.3)}$ | 81.8$_{(10.1)}$ | 68.8$_{(13.7)}$ | 88.3$_{(2.4)}$ | / | 1e7 |
| 6h vs 8z | Hard+ | **98.8**$_{(1.3)}$ | 93.8$_{(4.7)}$ | 88.4$_{(5.7)}$ | 0.3$_{(0.4)}$ | 9.7$_{(3.1)}$ | / | 1e7 |
| 3s5z vs 3s6z | Hard+ | **96.5**$_{(1.3)}$ | 85.3$_{(7.5)}$ | 84.3$_{(19.4)}$ | 82.8$_{(21.2)}$ | 68.8$_{(21.2)}$ | / | 2e7 |

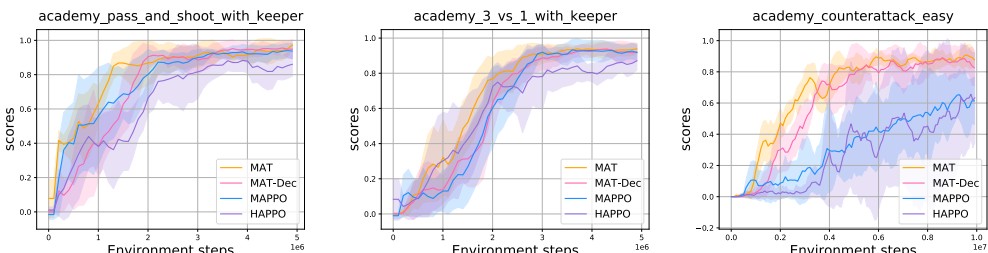

Figure 5: Performance comparison on the Google Research Football tasks with 2-4 agents from left to right respectively.

HAPPO on the time complexity. Lastly, to assure that the limiting joint policy is such that none of the agents is incentivized to change its policy (Nash equilibrium), MAT requires permutating the sequential order of updates at every iteration, which is inline with the discovery in HAPPO [15, Theorem 3].

## 5 Experiments and Results

MAT provides a new solution paradigm for cooperative MARL problems. The key insights of MAT are the sequential update scheme, which is inspired by Theorem (1), as well as the encoder-decoder architecture, which provides a highly-efficient implementation for a sequence modeling perspective. Importantly, MAT inherits the monotonic improvement guarantee, and agents' policies can be learned in parallel during training. We firmly believe MAT will become a game changer for MARL studies.

To evaluate if MAT meets our expectations, we test MAT on the StarCraftII Multi-Agent Challenge (SMAC) benchmark [31] where MAPPO with parameter sharing [46] has shown superior performance, and the Multi-Agent MuJoCo benchmark [7] where HAPPO [15] shows the current state-of-the-art performance. SMAC and MuJoCo environments are common benchmarks in the MARL field. On top of them, we also test MAT on the Bimanual Dexterous Hands Manipulation (Bi-DexHands) [6] which provides a list of challenging bimanual manipulation tasks (see Figure (3)), and the Google Research Football [18] benchmark with a series of cooperation scenarios in football game.

We apply the same hyper-parameters of baseline algorithms from their original paper to ensure their best performance, and adopt the same hyper-parameter tuning process for our methods with details in Appendix B. To ensure fair comparisons to CTDE methods, we also introduce a CTDE-variant of MAT called **MAT-Dec**, which essentially adopts a fully decentralized actor for each individual agent (rather than using the decoder proposed in MAT) while keeping the encoder fixed. The critic's

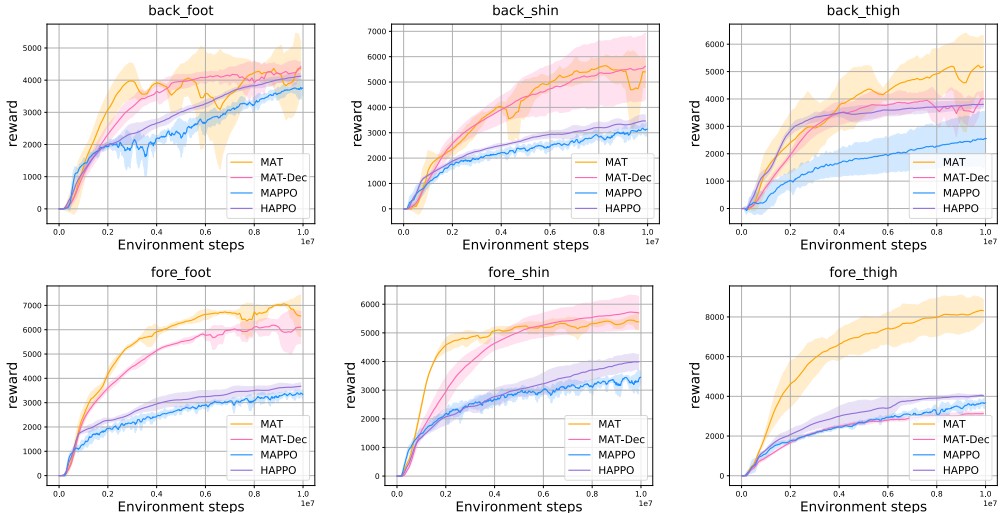

Figure 6: Performance on the HalfCheetah task with different disabled joints shown in Figure (3a).

loss for MAT-Dec is $L(\phi) = \frac{1}{T} \sum_{t=0}^{T-1} \left[ R(\mathbf{o}_t, \mathbf{a}_t) + \gamma \frac{1}{n} \sum_{m=1}^{n} V_{\bar{\phi}}(\hat{\mathbf{o}}_{t+1}^{i_m}) - \frac{1}{n} \sum_{m=1}^{n} V_{\phi}(\hat{\mathbf{o}}_t^{i_m}) \right]^2$, and we apply the local advantage estimation $A_t(\hat{\mathbf{o}}_t^{i_m}, a^{i_m})$ to guide the subsequent policy update.

## 5.1 Performance on Cooperative MARL Benchmarks

According to Table (1) and Figure (4), MAPPO significantly outperforms HAPPO in SMAC with higher sample efficiency. This verified the homogeneity of SMAC agents and the heterogeneity of multi-agent MuJoCo agents, that are also discovered by Kuba et al. [15]. Take the SMAC task *25m* as an example, all the marines are equivalent and interchangeable so that agents can learn from their teammate's experience. Sharing parameters in this settings means leveraging 25 times more examples to train each agents comparing with separated network of HAPPO, and thus enjoying higher learning efficiency. On the other hand, with the heterogeneous settings of multi-agent MuJoCo, training a "foot" agent with experience from a "thigh" agent can surely harm its performance since they represent different functions on the Cheetah. However, MAT outperforms MAPPO and HAPPO in almost all tasks in Table (1) and Figure (4), indicating its modeling capability on both homogeneous and heterogeneous-agent tasks. MAT also enjoys the superior performance over MAT-Dec, which emphasize the importance of the decoder architecture in the MAT design. On the Bi-DexHands tasks, MAT outperforms MAPPO and HAPPO methods by a large margin. We save the Google Football results to Figure (5), where the conclusion stays the same.

## 5.2 MAT as Excellent Few-short Learners

Since Transformer-based models often demonstrate strong generalization performance on few-short tasks [3, 9], we believe MAT can possess strong generalization ability on unseen MARL tasks as well. To validate such an assumption, we design zero-shot and few-shot experiments on SMAC and multi-agent MuJoCo tasks. For SMAC tasks, we pre-train agents on eight tasks involving five types of units (*3m*, *8m vs 9m*, *10m vs 11m*, *25m*, *3s vs 3z*, *2s3z*, *3s5z*, *MMM*) with 10M examples in total and then apply them on six separate and much harder tasks (*5m vs 6m*, *8m*, *27m vs 30m*, *2s vs 1sc*, *1c3s5z*, *MMM2*) including seven types of units. This setting is designed to evaluate the generalization ability of MAT when training on simple tasks but transferring to more diverse and complex downstream tasks. In terms of multi-agent MuJoCo, we reuse the models trained on the complete HalfCheetah robot as the pre-trained agent and then directly apply it to six new tasks, each with a different leg being disfunctioned (see Figure (3a)). We investigate the generalization capability of pre-trained models on each downstream task with 0% (zero-shot), 1%, 5%, 10% few-short new examples, respectively. Note that common MARL baselines such as HAPPO assume fixed number of agents during training, thus it cannot directly handle the cases with changing number of agents.

Table 2: Median evaluation win rate and the standard deviation on the SMAC benchmark for pre-trained models with different number of online examples.

| Methods | MAT | | | | MAPPO | | | | MAT-from scratch | | | |
|---|---|---|---|---|---|---|---|---|---|---|---|---|
| #examples | 0% | 1% | 5% | 10% | 0% | 1% | 5% | 10% | 0% | 1% | 5% | 10% |
| 5m vs 6m | $0.0_{(0.0)}$ | $0.0_{(0.0)}$ | $\mathbf{5.8}_{(3.1)}$ | $18.8_{(7.1)}$ | $0.0_{(0.0)}$ | $0.0_{(0.0)}$ | $4.3_{(3.8)}$ | $\mathbf{21.9}_{(12.2)}$ | $0.0_{(0.0)}$ | $0.0_{(0.0)}$ | $1.9_{(1.3)}$ | $3.8_{(2.1)}$ |
| 8m | $100_{(0.0)}$ | $100_{(1.2)}$ | $100_{(0.3)}$ | $100_{(2.1)}$ | $100_{(0.0)}$ | $100_{(1.4)}$ | $100_{(0.3)}$ | $100_{(1.4)}$ | $0.0_{(0.0)}$ | $10.6_{(23.8)}$ | $92.5_{(3.7)}$ | $100_{(1.4)}$ |
| 27m vs 30m | $0.0_{(0.0)}$ | $6.3_{(2.4)}$ | $\mathbf{53.8}_{(16.4)}$ | $\mathbf{71.2}_{(8.2)}$ | $9.4_{(3.6)}$ | $\mathbf{15}_{(5.9)}$ | $26.2_{(7.8)}$ | $26.8_{(9.7)}$ | $0.0_{(0.0)}$ | $0.0_{(0.0)}$ | $0.0_{(0.0)}$ | $0.3_{(15.6)}$ |
| 2s vs 1sc | $0.0_{(0.0)}$ | $15.6_{(13.8)}$ | $100_{(9.7)}$ | $100_{(0.0)}$ | $0.0_{(0.0)}$ | $\mathbf{43.1}_{(17.6)}$ | $100_{(1.1)}$ | $100_{(1.8)}$ | $0.0_{(0.0)}$ | $19.3_{(33.3)}$ | $96.3_{(6.2)}$ | $100_{(0.3)}$ |
| 1c3s5z | $\mathbf{3.1}_{(1.8)}$ | $\mathbf{5.6}_{(5.0)}$ | $\mathbf{82.5}_{(5.5)}$ | $\mathbf{100}_{(2.7)}$ | $\mathbf{3.1}_{(1.8)}$ | $4.3_{(4.9)}$ | $73.8_{(13.0)}$ | $97.5_{(2.1)}$ | $0.0_{(0.0)}$ | $7.5_{(4.8)}$ | $87.5_{(3.9)}$ | $100_{(1.4)}$ |
| MMM2 | $0.0_{(3.6)}$ | $0.0_{(1.8)}$ | $\mathbf{33.8}_{(13.7)}$ | $\mathbf{62.5}_{(12.1)}$ | $0.0_{(0.0)}$ | $0.0_{(1.4)}$ | $13.8_{(7.0)}$ | $36.2_{(9.6)}$ | $0.0_{(0.0)}$ | $0.0_{(0.0)}$ | $0.0_{(0.0)}$ | $0.0_{(0.7)}$ |

Table 3: Average evaluation score and standard deviation on Multi-Agent MuJoCo for pre-trained models with different number of online examples.

| Methods | MAT | | | | MAPPO | | | | MAT-from scratch | | | |
|---|---|---|---|---|---|---|---|---|---|---|---|---|
| #examples | 0% | 1% | 5% | 10% | 0% | 1% | 5% | 10% | 0% | 1% | 5% | 10% |
| back foot | $2100_{(89)}$ | $2837_{(95)}$ | $\mathbf{4691}_{(235)}$ | $\mathbf{5646}_{(79)}$ | $2936_{(301)}$ | $\mathbf{3017}_{(135)}$ | $3221_{(119)}$ | $3304_{(129)}$ | $-0.44_{(0.4)}$ | $-5.18_{(11)}$ | $670_{(1098)}$ | $1635_{(1184)}$ |
| back shin | $\mathbf{4005}_{(316)}$ | $\mathbf{4143}_{(230)}$ | $\mathbf{6077}_{(209)}$ | $\mathbf{7176}_{(74)}$ | $2406_{(32)}$ | $2542_{(108)}$ | $2796_{(137)}$ | $2955_{(127)}$ | $-0.31_{(0.1)}$ | $-3.95_{(17)}$ | $743_{(537)}$ | $1252_{(1123)}$ |
| back thigh | $\mathbf{5361}_{(45)}$ | $\mathbf{5641}_{(150)}$ | $\mathbf{7101}_{(119)}$ | $\mathbf{7460}_{(61)}$ | $3043_{(79)}$ | $3060_{(143)}$ | $3217_{(33)}$ | $3353_{(71)}$ | $-0.54_{(0.3)}$ | $-4.87_{(7.7)}$ | $930_{(589)}$ | $2067_{(861)}$ |
| fore foot | $\mathbf{1313}_{(512)}$ | $\mathbf{1955}_{(232)}$ | $\mathbf{4856}_{(146)}$ | $\mathbf{6054}_{(172)}$ | $623_{(44)}$ | $970_{(185)}$ | $2025_{(371)}$ | $2480_{(239)}$ | $-0.37_{(0.2)}$ | $-2.25_{(7.9)}$ | $1821_{(157)}$ | $2877_{(106)}$ |
| fore shin | $\mathbf{2435}_{(13)}$ | $\mathbf{2617}_{(71)}$ | $\mathbf{3851}_{(57)}$ | $\mathbf{4373}_{(83)}$ | $1715_{(55)}$ | $2457_{(125)}$ | $3096_{(59)}$ | $3310_{(54)}$ | $-0.15_{(0.06)}$ | $-0.96_{(6.0)}$ | $1461_{(101)}$ | $3003_{(316)}$ |
| fore thigh | $\mathbf{5631}_{(321)}$ | $\mathbf{6448}_{(417)}$ | $\mathbf{7952}_{(109)}$ | $\mathbf{8347}_{(81)}$ | $3087_{(110)}$ | $3171_{(83)}$ | $3340_{(52)}$ | $3519_{(59)}$ | $-0.29_{(0.3)}$ | $0.82_{(14)}$ | $1021_{(177)}$ | $2600_{(215)}$ |

We summarize the zero-shot and few-shot results of each algorithm in Table (2) and (3), where the bold number indicates the best performance. We also provide the performance of MAT if it was given the same amount of data but is trained from scratch, the "MAT-from scratch", as the control group to demonstrate the effectiveness of pre-training process. As both tables suggest, bold numbers are mainly located in the area of MAT, which justify MAT's strong generalisation performance as a few-short learner. Surprisingly, we find that the few-shot MAT with only 10% data show even higher rewards than its counterpart that is purely trained on HalfCheetah with the same disabled joints (*back foot*, *back shin* and *back thigh*) and 100% full amount data, we believe it is because the pre-train process offers initial weights that are not only closer to optimum but also less likely to stuck in bad local optima than random initialization.

## 6    Conclusion

In the past five years, large sequence models have achieved remarkable successes on solving visual language tasks. In this paper, we take the initial effort to build the connection between multi-agent reinforcement learning (MARL) problems and generic sequence models (SM), with the ambition that MARL researchers can hereafter benefit from the prosperous development on the sequence modeling side. Specifically, we contribute by unifying a general solution to cooperative MARL problems into a Transformer like encoder-decoder model. The proposed Multi-Agent Transformer (MAT) leverages the multi-agent advantage decomposition theorem, which essentially transforms the joint policy optimization process into a sequential decision making process that can be simply implemented by an auto-regressive model. We have demonstrated MAT's strong empirical performance on three challenging benchmarks against current state-of-the-art MARL solutions including MAPPO and HAPPO. Based on the established connection between MARL and SM, in the future, we plan to bring multi-agent learning tasks into large multi-modal SM, chasing for more generally intelligent models as the most recent success of GATO has already demonstrated [30].

## Acknowledgment

The SJTU team is partially supported by "New Generation of AI 2030" Major Project (2018AAA0100900), Shanghai Municipal Science and Technology Major Project (2021SHZDZX0102), Shanghai Sailing Program (21YF1421900), and National Natural Science Foundation of China (62076161, 62106141).

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
