# Appendices

## A  Algorithm Details

### A.1  Pseudo Code of Multi-Agent Transformer.

---
**Algorithm 1** Multi-Agent Transformer
---
1: **Input:** Stepsize $\alpha$, batch size $B$, number of agents $n$, episodes $K$, steps per episode $T$.
2: **Initialize:** Encoder $\{\phi_0\}$, Decoder $\{\theta_0\}$, Replay buffer $\mathcal{B}$.
3: **for** $k = 0, 1, \ldots, K - 1$ **do**
4:     **for** $t = 0, 1, \ldots, T - 1$ **do**
5:         Collect a sequence of observations $o_t^{i_1}, \ldots, o_t^{i_n}$ from environments.
        {// The Inference Phase}
6:         Generate representation sequence $\hat{o}_t^{i_1}, \ldots, \hat{o}_t^{i_n}$ by feeding observations to the encoder.
7:         Input $\hat{o}_t^{i_1}, \ldots, \hat{o}_t^{i_n}$ to the decoder.
8:         **for** $m = 0, 1, \ldots, n - 1$ **do**
9:           Input $a_t^{i_0}, \ldots, a_t^{i_m}$ and infer $a_t^{i_{m+1}}$ with the auto-regressive decoder.
10:        **end for**
11:       Execute joint actions $a_t^{i_0}, \ldots, a_t^{i_n}$ in environments and collect the reward $R(\boldsymbol{o}_t, \boldsymbol{a}_t)$.
12:       Insert $(\boldsymbol{o}_t, \boldsymbol{a}_t, R(\boldsymbol{o}_t, \boldsymbol{a}_t))$ in to $\mathcal{B}$.
13:     **end for**
    {// The Training Phase}
14:     Sample a random minibatch of $B$ steps from $\mathcal{B}$.
15:     Generate $V_\phi(\hat{o}^{i_1}), \ldots, V_\phi(\hat{o}^{i_n})$ with the output layer of the encoder.
16:     Calculate $L_{\text{Encoder}}(\phi)$ with Equation (4).
17:     Compute the joint advantage function $\hat{A}$ based on $V_\phi(\hat{o}^{i_1}), \ldots, V_\phi(\hat{o}^{i_n})$ with GAE.
18:     Input $\hat{o}^{i_1}, \ldots, \hat{o}^{i_n}$ and $a^{i_0}, \ldots, a^{i_{n-1}}$, generate $\pi_\theta^{i_1}, \ldots, \pi_\theta^{i_n}$ at once with the decoder.
19:     Calculate $L_{\text{Decoder}}(\theta)$ with Equation (5).
20:     Update the encoder and decoder by minimising $L_{\text{Encoder}}(\phi) + L_{\text{Decoder}}(\theta)$ with gradient descent.
21: **end for**

---

### A.2  Dynamic Process and Source Code of MAT

Please refer to https://sites.google.com/view/multi-agent-transformer

## B  Hyper-parameter Settings for Experiments

During experiments, the implementations of baseline methods are consistent with their official repositories, all hyper-parameters left unchanged at the origin best-performing status. The hyper-parameters adopted for different algorithms and tasks are listed in Table 4-12. In particular, the *ppo epochs* and *ppo clip* across different SMAC scenarios are unified to 10 and 0.05 respectively for pre-training and fine-tuning in few-shot experiments.

Table 4: Common hyper-parameters used for MAT, MAT-Dec, MAPPO and HAPPO in the SMAC domain.

| hyper-parameters | value | hyper-parameters | value | hyper-parameters | value |
|---|---|---|---|---|---|
| critic lr | 5e-4 | actor lr | 5e-4 | use gae | True |
| gain | 0.01 | optim eps | 1e-5 | batch size | 3200 |
| training threads | 16 | num mini-batch | 1 | rollout threads | 32 |
| entropy coef | 0.01 | max grad norm | 10 | episode length | 100 |
| optimizer | Adam | hidden layer dim | 64 | use huber loss | True |

Table 5: Different hyper-parameters used for MAT and MAT-Dec in the SMAC domain.

| maps | ppo epochs | ppo clip | num blocks | num heads | stacked frames | steps | $\gamma$ |
|---|---|---|---|---|---|---|---|
| 3m | 15 | 0.2 | 1 | 1 | 1 | 5e5 | 0.99 |
| 8m | 15 | 0.2 | 1 | 1 | 1 | 1e6 | 0.99 |
| 1c3s5z | 10 | 0.2 | 1 | 1 | 1 | 2e6 | 0.99 |
| MMM | 15 | 0.2 | 1 | 1 | 1 | 2e6 | 0.99 |
| 2c vs 64zg | 10 | 0.05 | 1 | 1 | 1 | 5e6 | 0.99 |
| 3s vs 5z | 15 | 0.05 | 1 | 1 | 4 | 5e6 | 0.99 |
| 3s5z | 10 | 0.05 | 1 | 1 | 1 | 3e6 | 0.99 |
| 5m vs 6m | 10 | 0.05 | 1 | 1 | 1 | 1e7 | 0.99 |
| 8m vs 9m | 10 | 0.05 | 1 | 1 | 1 | 5e6 | 0.99 |
| 10m vs 11m | 10 | 0.05 | 1 | 1 | 1 | 5e6 | 0.99 |
| 25m | 15 | 0.05 | 1 | 1 | 1 | 2e6 | 0.99 |
| 27m vs 30m | 5 | 0.2 | 1 | 1 | 1 | 1e7 | 0.99 |
| MMM2 | 5 | 0.05 | 1 | 1 | 1 | 1e7 | 0.99 |
| 6h vs 8z | 15 | 0.05 | 1 | 1 | 1 | 1e7 | 0.99 |
| 3s5z vs 3s6z | 5 | 0.05 | 1 | 1 | 1 | 2e7 | 0.99 |

Table 6: Different hyper-parameters used for MAPPO/HAPPO in the SMAC domain.

| maps | ppo epochs | ppo clip | hidden leyer | stacked frames | network | steps | $\gamma$ |
|---|---|---|---|---|---|---|---|
| 3m | 15 | 0.2 | 2 | 1 | rnn/mlp | 5e5 | 0.99/0.95 |
| 8m | 15 | 0.2 | 2 | 1 | rnn/mlp | 1e6 | 0.99/0.95 |
| 1c3s5z | 15 | 0.2 | 2 | 1 | rnn/mlp | 2e6 | 0.99/0.95 |
| MMM | 15 | 0.2 | 2 | 1 | rnn/mlp | 2e6 | 0.99/0.95 |
| 2c vs 64zg | 5 | 0.2 | 2 | 1 | rnn/mlp | 5e6 | 0.99/0.95 |
| 3s vs 5z | 15 | 0.05 | 2 | 4 | mlp | 5e6 | 0.99/0.95 |
| 3s5z | 5 | 0.2 | 2 | 1 | rnn/mlp | 3e6 | 0.99/0.95 |
| 5m vs 6m | 10 | 0.05 | 2 | 1 | rnn/mlp | 1e7 | 0.99/0.95 |
| 8m vs 9m | 15 | 0.05 | 2 | 1 | rnn/mlp | 5e6 | 0.99/0.95 |
| 10m vs 11m | 10 | 0.2 | 2 | 1 | rnn/mlp | 5e6 | 0.99/0.95 |
| 25m | 10/5 | 0.2 | 2/1 | 1 | rnn/mlp | 2e6 | 0.99/0.95 |
| 27m vs 30m | 5 | 0.2 | 2/1 | 1 | rnn/mlp | 1e7 | 0.99/0.95 |
| MMM2 | 5 | 0.2 | 2/1 | 1 | rnn/mlp | 1e7 | 0.99/0.95 |
| 6h vs 8z | 5 | 0.2 | 2/1 | 1 | mlp | 1e7 | 0.99/0.95 |
| 3s5z vs 3s6z | 5 | 0.2 | 2 | 1 | mlp | 2e7 | 0.99/0.95 |

Table 7: Common hyper-parameters used for all methods in the multi-agent MuJoCo domain.

| hyper-parameters | value | hyper-parameters | value | hyper-parameters | value |
|---|---|---|---|---|---|
| gamma | 0.99 | steps | 1e7 | stacked frames | 1 |
| gain | 0.01 | optim eps | 1e-5 | batch size | 4000 |
| training threads | 16 | num mini-batch | 40 | rollout threads | 40 |
| entropy coef | 0.001 | max grad norm | 0.5 | episode length | 100 |
| optimizer | Adam | hidden layer dim | 64 | use huber loss | True |

Table 8: Different hyper-parameter used in the mulit-agent MuJoCo domain.

| hyper-parameters | MAT | MAT-Dec | MAPPO | HAPPO |
|---|---|---|---|---|
| critic lr | 5e-5 | 5e-5 | 5e-3 | 5e-3 |
| actor lr | 5e-5 | 5e-5 | 5e-6 | 5e-6 |
| ppo epochs | 10 | 10 | 5 | 5 |
| ppo clip | 0.05 | 0.05 | 0.2 | 0.2 |
| num hidden layer | / | / | 2 | 2 |
| num blocks | 1 | 1 | / | / |
| num head | 1 | 1 | / | / |

Table 9: Common hyper-parameters used for all methods in the Bi-DexHands domain.

| hyper-parameters | value | hyper-parameters | value | hyper-parameters | value |
|---|---|---|---|---|---|
| gamma | 0.96 | steps | 5e7 | stacked frames | 1 |
| gain | 0.01 | optim eps | 1e-5 | ppo epochs | 5 |
| ppo clip | 0.2 | num mini-batch | 1 | rollout threads | 80 |
| batch size | 6000 | episode length | 75 | optimizer | Adam |
| entropy coef | 0.001 | max grad norm | 0.5 | training threads | 16 |

Table 10: Different hyper-parameter used in the Bi-DexHands domain.

| hyper-parameters | MAT | MAT-Dec | MAPPO | HAPPO |
|---|---|---|---|---|
| critic lr | 5e-5 | 5e-5 | 5e-4 | 5e-4 |
| actor lr | 5e-5 | 5e-5 | 5e-4 | 5e-4 |
| hidden dim | 64 | 64 | 512 | 512 |
| num hidden layer | / | / | 1 | 1 |
| num blocks | 1 | 1 | / | / |
| num head | 1 | 1 | / | / |

Table 11: Common hyper-parameters used for all methods in the Google Research Football domain.

| hyper-parameters | value | hyper-parameters | value | hyper-parameters | value |
|---|---|---|---|---|---|
| critic lr | 5e-4 | actor lr | 5e-4 | gamma | 0.99 |
| gain | 0.01 | optim eps | 1e-5 | batch size | 4000 |
| training threads | 16 | num mini-batch | 1 | rollout threads | 20 |
| entropy coef | 0.01 | max grad norm | 0.5 | episode length | 200 |
| optimizer | Adam | hidden layer dim | 64 | stacked frames | 1 |

Table 12: Different hyper-parameter used in the Google Research Football domain.

| hyper-parameters | MAT | MAT-Dec | MAPPO | HAPPO |
|---|---|---|---|---|
| ppo epochs | 10 | 10 | 5 | 5 |
| ppo clip | 0.05 | 0.05 | 0.2 | 0.2 |
| num hidden layer | / | / | 2 | 2 |
| num blocks | 1 | 1 | / | / |
| num head | 1 | 1 | / | / |

# C   Details of Experimental Results

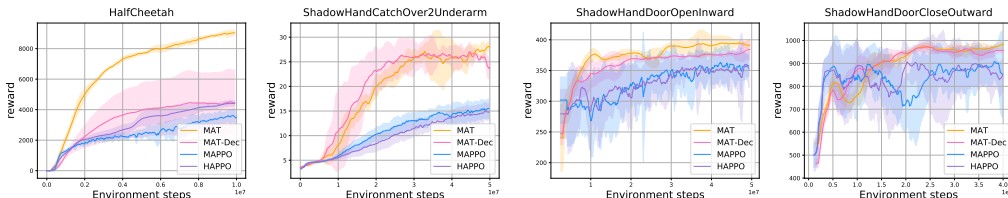

Figure 7: Performance comparisons on the Multi-Agent MuJoCo and the Bi-DexHands benchmarks, showing MAT's advantages in robot control.

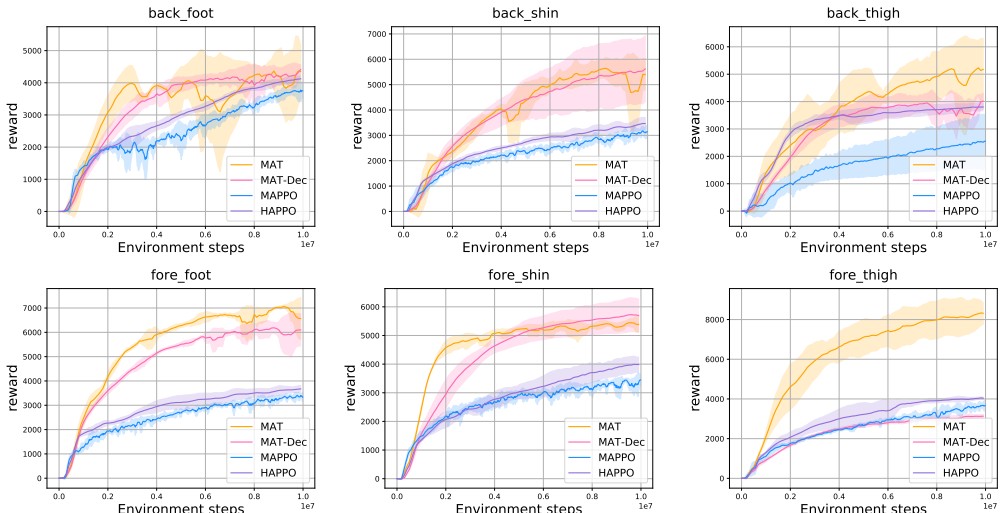

Figure 8: Performance comparison on the HalfCheetah tasks with different disabled joints, where MAT significantly outperformed baseline methods. Together with performance on the complete HalfCheetah in Figure (7), it emphasises MAT's capability for heterogeneous-agent tasks (agents are not interchangeable).

Table 13: Performance evaluations of win rate and standard deviation on the SMAC benchmark, where UPDeT's official codebase supports several Marine-based tasks only.

| Task | Difficulty | MAT | MAT-Dec | MAPPO | HAPPO | QMIX | UPDeT | Steps |
|------|-----------|-----|---------|-------|-------|------|-------|-------|
| 3m | Easy | **100.0**$_{(1.8)}$ | **100.0**$_{(1.1)}$ | **100.0**$_{(0.4)}$ | **100.0**$_{(1.2)}$ | 96.9$_{1.3}$ | **100.0**$_{(5.2)}$ | 5e5 |
| 8m | Easy | **100.0**$_{(1.1)}$ | 97.5$_{(2.5)}$ | 96.8$_{(2.9)}$ | 97.5$_{(1.1)}$ | 97.7$_{1.9}$ | 96.3$_{(9.7)}$ | 1e6 |
| 1c3s5z | Easy | **100.0**$_{(2.4)}$ | **100.0**$_{(0.4)}$ | **100.0**$_{(2.2)}$ | 97.5$_{(1.8)}$ | 96.9$_{(1.5)}$ | / | 2e6 |
| MMM | Easy | **100.0**$_{(2.2)}$ | 98.1$_{(2.1)}$ | 95.6$_{(4.5)}$ | 81.2$_{(22.9)}$ | 91.2$_{(3.2)}$ | / | 2e6 |
| 2c vs 64zg | Hard | **100.0**$_{(1.3)}$ | 95.9$_{(2.3)}$ | **100.0**$_{(2.7)}$ | 90.0$_{(4.8)}$ | 90.3$_{(4.0)}$ | / | 5e6 |
| 3s vs 5z | Hard | **100.0**$_{(1.7)}$ | **100.0**$_{(1.3)}$ | **100.0**$_{(2.5)}$ | 91.9$_{(5.3)}$ | 92.3$_{(4.4)}$ | / | 5e6 |
| 3s5z | Hard | **100.0**$_{(1.9)}$ | **100.0**$_{(3.3)}$ | 72.5$_{(26.5)}$ | 90.0$_{(3.5)}$ | 84.3$_{(5.4)}$ | / | 3e6 |
| 5m vs 6m | Hard | **90.6**$_{(4.4)}$ | 83.1$_{(4.6)}$ | 88.2$_{(6.2)}$ | 73.8$_{(4.4)}$ | 75.8$_{(3.7)}$ | **90.6**$_{(6.1)}$ | 1e7 |
| 8m vs 9m | Hard | **100.0**$_{(3.1)}$ | 95.0$_{(4.6)}$ | 93.8$_{(3.5)}$ | 86.2$_{(4.4)}$ | 92.6$_{(4.0)}$ | / | 5e6 |
| 10m vs 11m | Hard | **100.0**$_{(1.4)}$ | **100.0**$_{(2.0)}$ | 96.3$_{(5.8)}$ | 77.5$_{(9.7)}$ | 95.8$_{(6.1)}$ | / | 5e6 |
| 25m | Hard | **100.0**$_{(1.3)}$ | 86.9$_{(5.6)}$ | **100.0**$_{(2.7)}$ | 70.1$_{(8.1)}$ | 90.2$_{(9.8)}$ | 2.8$_{(3.1)}$ | 2e6 |
| 27m vs 30m | Hard+ | **100.0**$_{(0.7)}$ | 95.3$_{(2.2)}$ | 93.1$_{(3.2)}$ | 5.6$_{(2.8)}$ | 39.2$_{(8.8)}$ | / | 1e7 |
| MMM2 | Hard+ | **93.8**$_{(2.6)}$ | 91.2$_{(5.3)}$ | 81.8$_{(10.1)}$ | 68.8$_{(13.7)}$ | 88.3$_{(2.4)}$ | / | 1e7 |
| 6h vs 8z | Hard+ | **98.8**$_{(1.3)}$ | 93.8$_{(4.7)}$ | 88.4$_{(5.7)}$ | 0.3$_{(0.4)}$ | 9.7$_{(3.1)}$ | / | 1e7 |
| 3s5z vs 3s6z | Hard+ | **96.5**$_{(1.3)}$ | 85.3$_{(7.5)}$ | 84.3$_{(19.4)}$ | 82.8$_{(21.2)}$ | 68.8$_{(21.2)}$ | / | 2e7 |

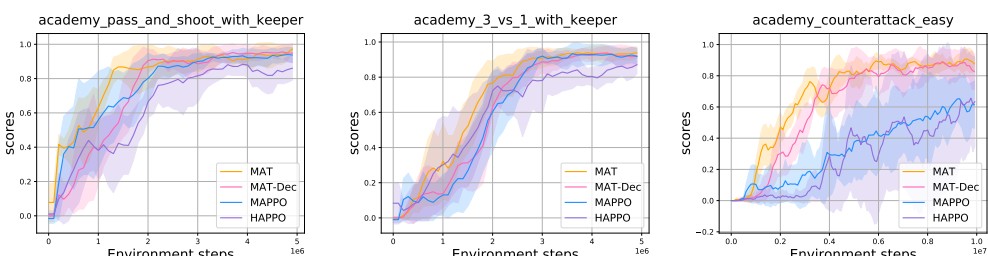

Figure 9: Performance comparison on SMAC tasks. MAT consistently outperforms its rivals, indicating its modeling capability for homogeneous-agent tasks (agents are interchangeable).

Figure 10: Performance comparison on the Google Research Football tasks with 2-4 agents from left to right respectively, telling the same conclusion that MAT outperforms MAPPO and HAPPO.

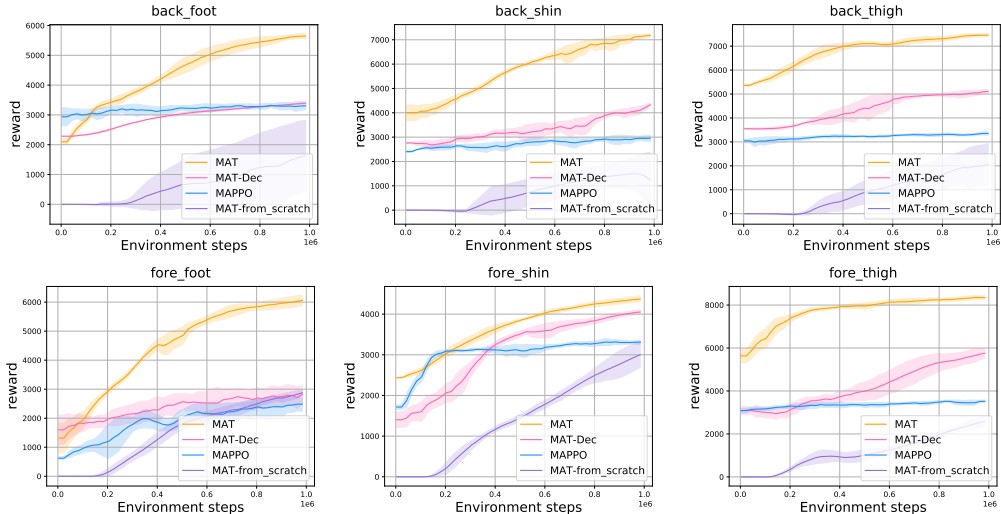

Figure 11: Few-shot performance comparison with pre-trained models on multi-agent MuJoCo tasks. MAT exhibits powerful generalisation capability when parts of the robot fail.

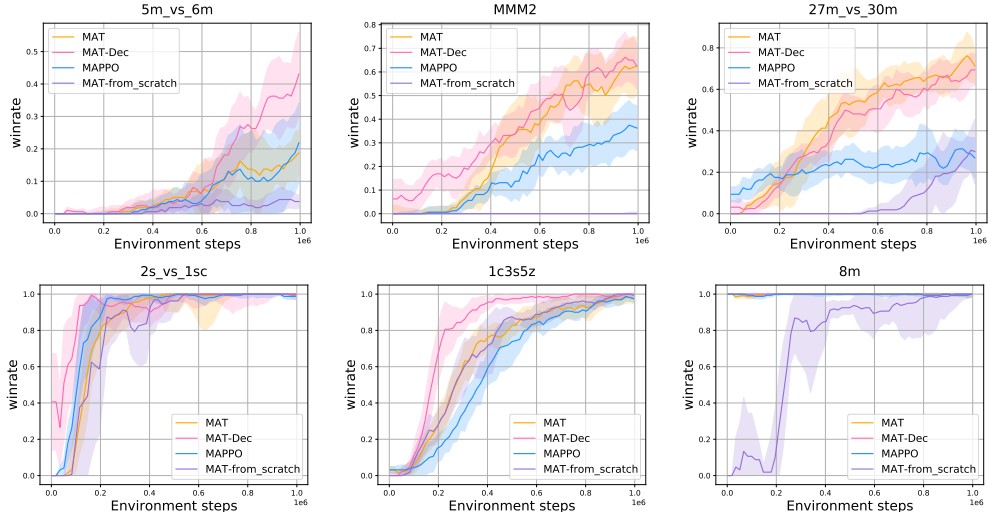

Figure 12: Few-shot performance comparison with pre-trained models on SMAC tasks. Sequence-modeling-based methods, MAT and MAT-Dec, enjoy superior performance over MAPPO, justifying their strong generalisation capability as few-shot learners.

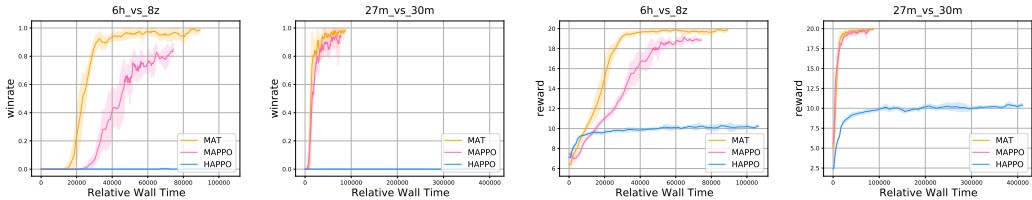

Figure 13: Wall clock time comparison on scenarios with 10M environment steps and different size of agents, demonstrating that MAT enjoys better computational efficiency than HAPPO, especially with large number of agents. Since the winrate of HAPPO is close to zero in these super-hard settings, we present their episodic rewards as well for more details.

# D Ablation studies

In this section, we conduct ablation studies to investigate the importance of different components. Since agents are shuffled at every iteration, the position encoding in vanilla Transformer might not be very relevant in MARL tasks, and has been already discarded in our implementation. Instead, we bind each observation with corresponding one-hot agent id and build an ablation experiment to investigate their effect. The results in Figure (14) confirmed our choice, where the agent id encoding significantly outperforms position encoding with regard to episodic rewards and stability. Further, applying position encoding on top of agent id encoding can not enjoy extra performance boost.

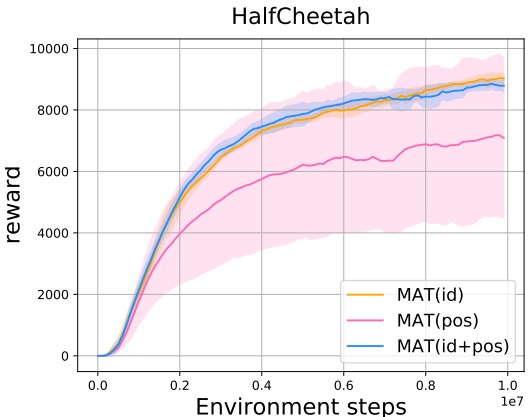

Figure 14: Performance comparison for different observation encoding approaches on a heterogeneous scenario, i.e. HalfCheetah, where MAT(id) encodes inputs with one-hot agent id, which is applied in our implementation; MAT(pos) encodes inputs with their positions in sequences; MAT(id+pos) encodes inputs with both their agent id and positions.

Besides, we also compared the implementation with different model architectures, i.e. encoder-decoder, encoder only, decoder only, and other sequence models (GRU) to verify the necessity of different architecture components. We build this ablation on both the homogeneous and heterogeneous scenarios as shown in Figure (15), where the complete Transformer architecture achieves the best performance, emphasizing the advantage of the Transformer and the necessity of encoder-decoder architectures.

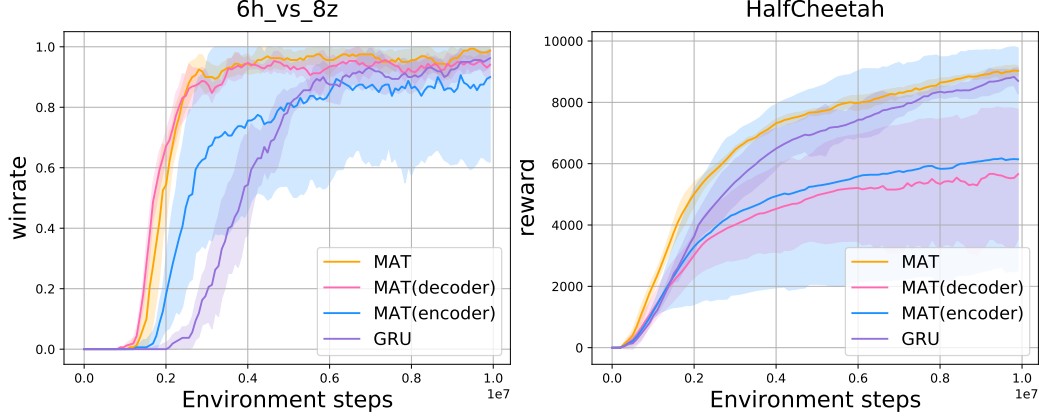

Figure 15: Performance comparison for different model architectures to explore the effect of each component, where MAT is the the original implementation; MAT(decoder) is implemented with the decoder only, keeping the auto-regressive process; MAT(encoder) is implemented with the encoder only, without the the auto-regressive process; the GRU maintains the encoding and decoding process but implements them with GRU networks.