# OpenReview forum: "Multi-Agent Reinforcement Learning is a Sequence Modeling Problem"
_NeurIPS.cc/2022/Conference — NeurIPS 2022 Accept_

### Official Review · Reviewer_rnUZ · 2022-07-03

**Rating:** 6
**Confidence:** 4
**Soundness:** 2 fair
**Presentation:** 4 excellent
**Contribution:** 3 good

**Summary:**

The paper introduces a sequence model problem formulation of the cooperative MARL setup. Multi-Agent Transformer (MAT) is proposed to extend HAPPO by an encoder-decoder architecture which maps sequences of agent observations to sequences of agent actions. MAT is evaluated in SMAC, MA-Mujoco, and Dexterous Hands Manipulation to demonstrate superior performance over MAPPO and HAPPO, generalization, and few-shot learning capabilities.

**Questions:**

According to the SMAC documentation, the action spaces vary depending on the number of enemies. How does MAT deal with that in the experiments?

**Limitations:**

Limitations and potential negative societal impact have not been addressed in this paper.

**Strengths And Weaknesses:**

- **Originality**: The paper presents an interesting and novel direction in cooperative multi-agent learning using state-of-the-art sequence-to-sequence learning techniques.
- **Quality**: The paper is mostly technically sound. The explanations and experiments are presented well and the results adequately support the claims. However, the problem formulation in Section 2.1 actually represents a *Markov game* rather than a Dec-POMDP (according to [1] and the HAPPO paper). Please note that unlike the (unknown) state, the joint observation is not Markovian in Dec-POMDPs. Therefore, the value functions defined in Eq. 1 are only valid for Markov games but not for Dec-POMDPs [2].
- **Clarity**: The paper is very well-written, clearly structured, and easy to follow.
- **Significance**: The experimental results indicate a clear improvement over MAPPO and HAPPO with respect to performance, generalization, and few-shot learning capabilities. Due to the parallelizability of transformers, MAT potentially offers better computational efficiency than HAPPO although a report on wall clock time would have been nice.

Minor comments:
- Text and labels in Fig. 4 are very small and hard to read when printed. Labels in Fig. 5 are fine but the axes ticks are too small (I suggest to reduce the number of ticks and make the text larger).
- The numbers in brackets in Table 2 and 3 are unreadable when printed.

[1] Littmann, "Markov Games as a Framework for Multi-Agent Reinforcement Learning", ICML 1994

[2] Oliehoek et al., "Optimal and Approximate Q-value Functions for Decentralized POMDPs", JAIR 2008

---

> ### Author Response · Authors · 2022-08-02
> **Response to Reviewer rnUZ**
>
> ### We thank Reviewer rnUZ for his/her constructive comments that will surely turn our paper into a better shape.
> > **Q1** The problem formulation is that of Markov games [1] rather than the stated Dec-POMDPs. The HAPPO paper uses the *Markov game* formulation as well, thus the deployed theoretical results are sound in this setting only. This should be corrected.
>
> **A1** The reviewr is right in pointing this out. We apologize for this error. We had changed *decentralized Partially Observable Markov Decision Processes (Dec-POMDPs)* to *Markov Games (MGs)* in the revision and keep the stated formulation.
>
> > **Q2** MAT achieves computational-efficiency advantage w.r.t. HAPPO. It would be nice to see the exact performance comparison w.r.t. wall clock time.
>
> **A2** Thanks so much for the suggestion, the wall clock time comparison between MAT, MAPPO and HAPPO w.r.t 10M environment steps is demonstrated under this **anonymized** [link](https://anonymous.4open.science/r/MAT_Rebuttal-0ABA/walltime.pdf). Also, we had added this comparison in our revised Appendix to better demonstrate the advantages.
>
> > **Q3** Figures 4 & 5, as well Tables 2 & 3 are illegible due to size.
>
> **A3** This is a valuable feedback. With the suggestion in mind, we will readjust the style of all figures and tables to make them more readable in the final version.
>
> **Reference:**
> [1] Littmann, "Markov Games as a Framework for Multi-Agent Reinforcement Learning", ICML 1994

---

### Official Review · Reviewer_MEiB · 2022-07-11

**Rating:** 7
**Confidence:** 4
**Soundness:** 3 good
**Presentation:** 3 good
**Contribution:** 2 fair

**Summary:**

This paper studies transformer networks in an actor-critic algorithm with a multi-agent setting. The authors first picks a random order of agents, motivated by the advantage decomposition theorem. Given this ordering of agents and their observations and actions, they formulate multi-agent RL as a sequence modeling problem. A transformer encoder is first applied to a sequence of observations to generate observation encodings and value of each agent. Conditioned on observation encodings, a transformer decoder is applied to a sequence of actions to generate action distributions of every agent. A new ordering of agents is resampled for every iteration to prevent agents from changing their policy. Different from HAPPO, the authors use a different objective where parameters of all agents are updated simultaneously rather than sequentially, which can potentially be faster with gradient accumulation and batching. When evaluated on a set of cooperative multi-agent environments, the proposed model, MAT, outperforms strong baselines. It is also shown to perform well with scarce data.

**Questions:**

I have several questions regarding my concerns above.

1. Can you compare MAT to other sequential models, especially ones that use transformer architecture? If there are comparable approaches, can you add baseline experiments?
2. Can you clarify the advantage of a transformer decoder w.r.t. parallel training?
3. Can you add ablation studies to understand different components in the transformer? Also, another ablation on just using the decoder where observations and actions for all agents $i_{1:m-1}$ are provided as input to the decoder to generate value and action distribution for agent $i_m$? This would be useful to understand if the added complexity of encoder-decoder architecture, which is inspired by the decomposition theorem where observations of all agents are given, is needed to get improved results.

**Limitations:**

The authors addressed the limitations and included a source code with their submission.

**Strengths And Weaknesses:**

The main strength is improved performance and shared experience from all agents in a flexible framework like transformers. Main weaknesses are lack of related work and some of the claims.

## Strengths
1. The paper is overall well written and easy to follow.
2. The authors use a transformer architecture in a multi-agent setting and creates a connection to the multi-agent advantage decomposition theorem.
3. Experimental results show improvements.

## Weaknesses
1. The main contribution of the paper is the transformer architecture but related work lacks any prior work on using sequential models in a multi-agent setting. It is not easy to position the paper without knowing if the encoder-decoder structure, transformers, or advantage decomposition has been studied in the past. Especially, UPDeT*, which also uses a transformer architecture and a similar encoder-decoder structure, should be compared.
2. You mention that "the transformer architecture allows sequential policies to be trained in parallel" but it is not clear what this means. While you are using a different objective with all actions of all agents are available, you still need to run the decoder on $\hat{a}^{i_{1:m-1}}_t$ to generate the action distribution for agent $i_m$. Main difference from the autoregressive variant seems to be batching of input and gradient accumulation but not clear if this is what you meant by parallelism.
3. It is not clear if all of transformer components are useful in this setting. For example, since you are shuffling agents at every iteration, position encoding might not be very relevant and can be discarded. I think an ablation study is needed to understand the effect of different components.


\* UPDET: UNIVERSAL MULTI-AGENT REINFORCEMENT LEARNING VIA POLICY DECOUPLING WITH TRANSFORMERS. Siyi Hu, Fengda Zhu, Xiaojun Chang, Xiaodan Liang.

---

> ### Author Response · Authors · 2022-08-02
> **Response to Reviewer MEiB**
>
> ### We thank Reviewer MEiB for his/her constructive comments that will surely turn our paper into a better shape.
> > **Q1** The work lacks a *related works* section which makes it difficult to position the used techniques (advantage decomposition, transformers) in the MARL literature. The UPDeT algorithm, which also uses transformers, is not compared against.
>
> **A1** We appreciate pointing out this flaw. The Policy Decoupling in UPDeT is relevant to our work. UPDeT focuses on observations of each single agent, handling various observation sizes by decoupling the observations into a set of observation-entities according to the physical meanings, matching them with different action-groups, and modeling the relationship between the matched observation-entities with self-attention mechanism for better representation learning [1]. In contrast, MAT focuses more on observations and actions from different agents, modeling the interrelationship between agents with Transformer architectures, promoting the monotonic improvement of joint policies without the requirement for divisible observations.
>
> Besides, Section 1 was intended to provide the discussion about the related work and the motivation and blend them together, whereas Section 2 continued the literature further mathematically by covering the related work (such as the trust-region methods, e.g., MAPPO and HAPPO) that is closely related to the advantage decomposition theorem. That said, thanks for the reviewer’s comment, in the updated version, we have better positioned our work in the general area of MARL by:
>
> 1. adding the discussion about two more classic MARL methods, i.e. QMIX (value decomposition) and MADDPG (deterministic policy gradient), and a Transformer-based MARL method, i.e. UPDeT, in Sections 1 and 2.
> 2. Additionally, running more baseline experiments for MADDPG and UPDeT, where currently completed results are shown in this **anonymous** [link](https://anonymous.4open.science/r/MAT_Rebuttal-0ABA/baseline.pdf) (QMIX’s performance is covered already in Tab.1).
>
>
> > **Q2** You mention that "the transformer architecture allows sequential policies to be trained in parallel" but it is not clear what this means. If you refer to parallelism in the agents' updates, then it comes as counterintuitive. After all, you need to run the decoder on $\hat{a}^{i_{1:m-1}}_t$ to generate the action distribution for agent $i_m$.
>
> **A2** We apologize for the confusion. What we want to say here is that "In order to guarantee the monotonic improvement of joint policies, HAPPO has to update each policy one-by-one during training, by leveraging previous update results of $\pi^{i_1},...,\pi^{i_{m-1}}$ to improve $\pi^{i_m}$, which becomes critical at large size of agents. By contrast, the attention mechanism of Transformer architecture (especially the decoder) allows for batching the ground truth actions $a_t^{i_0},...,a_t^{i_{n-1}}$ in the buffer to predict $a_t^{i_1},...,a_t^{i_{n}}$ and update policies simultaneously, which significantly improves the training speed and makes it feasible for large size of agents". We had added this clarification to the last paragraph of Section 3 in our revision.
>
>
> > **Q3** Ablation studies could be helpful to understand the effect of different components. For example, since agents are shuffled at every iteration, the position encoding might not be very relevant and can be discarded. Another example is to use the decoder where observations and actions for all agents $i_{1:m-1}$ are provided as input to the decoder to generate value and action distribution for agent $i_m$.
>
> **A3**
> We are very grateful for the insightful suggestion and have conducted more ablation experiments w.r.t different encoding approaches and model architectures, as described at the end of revised appendices.
>
> Exactly as you mentioned, we found the position encoding in the vanilla Transformer [1] is not very relevant for MARL tasks in early experiments, and it has been already discarded in our implementation. Instead, we bind each observation with corresponding one-hot agent id, which also be applied to baseline methods for fairness. The ablation results under this **anonymous** [link](https://anonymous.4open.science/r/MAT_Rebuttal-0ABA/encoding.pdf) emphasized the importance of agent id encoding on heterogeneous settings.
>
> With the suggestion in mind, we also compared the implementation with different model architectures, i.e. encoder-decoder, encoder only, decoder only, and other sequence models (GRU). Corresponding ablation results under this **anonymous** [link](https://anonymous.4open.science/r/MAT_Rebuttal-0ABA/structure.pdf) confirmed the advantage of Transformer and the necessity of encoder-decoder architectures.
>
> **Reference:**
>
> [1] Ashish Vaswani, Noam Shazeer, Niki Parmar, Jakob Uszkoreit, Llion Jones, Aidan N Gomez, Łukasz Kaiser, and Illia Polosukhin. Attention is all you need. In Advances in neural information processing systems, pages 5998–6008, (2017)

---

> > ### Comment · Reviewer_MEiB · 2022-08-09
> > **Response to Authors**
> >
> > Thank you for the clarification. I would be increasing my score.
> >
> > It would help if you can explain the difference between UPDeT and yours better. For example, UPDeT also uses transformers to model interrelationships between agents.

---

> > > ### Author Response · Authors · 2022-08-10
> > > **Thanks for upgrading the score**
> > >
> > > We would like to say thank you for your deep engagement with our work, especially the direction you have suggested helps us make the paper more understandable. Thanks a lot.
> > >
> > > Indeed, both UPDeT and MAT are trying to model the interrelationships between agents using Transformers. However, while UPDeT implements it by modeling the relationship between *matched observation-entities* within each agent's local observations and action spaces, MAT implements it by modeling the relationship between different agents' observations and actions directly and enjoys the advantages in the following three aspects:
> > > 1. MAT does not require observations to be *divisible* and thus is applicable for more scenarios. UPDeT needs to divide each observation into a set of *observation-entities* (each *observation-entity* contains information about an ally or enemy unit) before modeling, requiring each observation contains information of all units as well as is well-structured, i.e. *divisible*, and limiting its application. For example, considering scenarios where each observation does not contain information about all units, like Multi-agent MuJoCo whose observations involve only a current controlled joint and possibly some neighbor joints (sometimes they even contain only a single joint so that is hard to be divided), UPDeT might not be able to model the interrelationships between all agents without extra works to reorganize observations. By contrast, MAT models different agents' observations directly, and thus no assumptions are required about the content and structure of observations.
> > > 2. MAT supports both discrete and continuous action spaces, while UPDeT supports only discrete action spaces that could be divided into a set of *action-groups*.
> > > 3. MAT provides an explicit guarantee for monotonic improvement of joint policies since it considers the relationship between actions of different agents with the auto-regressive decoder, while UPDeT still risks conflicting joint actions because it considers only the relationship between *agent-groups* of a single agent.

---

### Official Review · Reviewer_YpeW · 2022-07-11

**Rating:** 6
**Confidence:** 3
**Soundness:** 3 good
**Presentation:** 3 good
**Contribution:** 3 good

**Summary:**

This work proposes the use of sequence models (SMs), specifically Transformers, to implement the Multi-Agent Advantage Decomposition theorem. This is done by introducing the Multi-Agent Transformer (MAT), an encoder-decoder model that sequentially predicts the action of an agent based the actions taken by previous agents in a random permutation. MAT is evaluated on four different benchmarks: SMAC, Multi-Agent MuJoCo benchmark, Bi-DexHands and Google Research Football benchmark. MAT achieves strong performance on SMAC and MuJoCo compared to the provided baselines. Further experiments show evidence that MAT can be good at few-shot learning.

**Questions:**

- This work seems relevant and should probably be addressed and possibly compared to:
[1] Hu, Siyi, Fengda Zhu, Xiaojun Chang, and Xiaodan Liang. "Updet: Universal multi-agent reinforcement learning via policy decoupling with transformers." arXiv preprint arXiv:2101.08001 (2021).

- What would happen to its performance if we replaced the transformer by other types of recurrent models (GRU, LSTM)?


**Limitations:**

Limitations were not addressed in the manuscript, as reported by the authors.

**Strengths And Weaknesses:**

Strengths
- I found the application of Transformers’ causal masked attention to the Multi-Agent Advantage Decomposition theorem novel.
- The performed experiments were instructive and provide good insight into the MAT’s performance. I particularly appreciated it’s ability to model both homogeneous and heterogenous agent tasks.
- I thought the paper was well organized and well-written.

Weaknesses:
- Where is the related work section? This is an important part of any scientific paper as it positions this work in the general area, (i.e., MARL). Only covering two contemporary methods is not sufficient in my opinion.
- I found a MAPPO result mismatch with paper; for example, for MMM2, the original paper reports 90.6 win rate versus 81.8 in the current manuscript. Did the authors implement their own version? If so, why not use the published version of the results?
- I found the explanation of the few-shot experiments confusing and unclear (section 5.2). For instance what exactly was done in the last meta-column (MAT-from scratch)? Also to help with the flow, I would suggest moving the paragraph starting at line 265 to the beginning of section 5.2.

---

> ### Author Response · Authors · 2022-08-02
> **Response to Reviewer YpeW: Part I**
>
> ### We thank Reviewer YpeW for his/her constructive comments that will surely turn our paper into a better shape.
> > **Q1** Why have the authors not include a *related works* section and only reviewed two trust-region mehtods? *Related works* serves an important role of positioning the paper in the MARL literature.
>
> **A1** Section 1 was intended to provide the discussion about the related work and the motivation and blend them together, whereas Section 2 continued the literature further mathematically by covering the related work (such as the trust-region methods, e.g., MAPPO and HAPPO) that is closely related to the advantage decomposition theorem.  That said, thanks for the reviewer’s comment, in the updated version, we have better positioned our work in the general area of MARL by:
>
> 1. adding the discussion about two more classic MARL methods, i.e. QMIX (value decomposition) and MADDPG (deterministic policy gradient), and a Transformer-based MARL method, i.e. UPDeT, in Section 1 and 2.
> 2. Additionally, running more baseline experiments for MADDPG and UPDeT, where currently completed results are shown in this ***anonymous*** [link](https://anonymous.4open.science/r/MAT_Rebuttal-0ABA/baseline.pdf) (QMIX’s performance is covered already in Tab.1).
>
>
> > **Q2** There seems to be a mismatch between the MAPPO performance from the original paper [3] and the one reported in this work. For example, on MMM2, MAPPO is known to achieve 90.6 win rate, while here authors report 81.8. Did the authors implement their own MAPPO version? If so, why?
>
> **A2** We used the original MAPPO implementation from the MAPPO codebase. Also, we kept its hyperparameters unchanged. What seems to be the reason of the mismatch is that we ran MAPPO with seeds generated by us at random (MAPPO paper has not listed the details of their seeds). Therefore, random seeds are different between our paper and MAPPO paper. The striking difference comes from the fact that MAPPO performed particularly poorly with one of the seeds, which is reflected by the higher variance in Tab.1 (10.1 versus 2.8). For better demonstration, we here provide the plot after removing the outlier under the following ***anonymised*** [link](https://anonymous.4open.science/r/MAT_Rebuttal-0ABA/outlier.pdf), where MAPPO achieve 90.2 win rate. Moreover, MAT achieves 93.8 win rate on MMM2, which is still better than the known win rate of MAPPO.
>
>
> > **Q3** Some parts of explanation of the few-shot experiments (Section 5.2) are confusing. For instance what exactly was done in the last meta-column (MAT-from scratch)? Also to help with the flow, I would suggest moving the paragraph starting at line 265 to the beginning of section 5.2.
>
> **A3** We apologize for the confusion. The difference between MAT and "MAT-from scratch" in section 5.2 is that the MAT model is pre-trained on different tasks while the "MAT-from scratch" model is randomly initialized. Therefore, the "MAT-from scratch" is a control group experiment to demonstrate the effectiveness of pre-training process w.r.t MAT. Also, with reviewer's suggestion in mind, we have moved the paragraph mentioned to section 5.2 as well as clarified the "MAT-from scratch" in our revision since it does make our paper easier to follow.

---

> > ### Author Response · Authors · 2022-08-02
> > **Response to Reviewer YpeW: Part II**
> >
> > >**Q4** This work seems relevant and should probably be addressed and possibly compared to UPDeT [1].
> >
> > **A4** We agree that the Policy Decoupling in UPDeT is relevant to our work. UPDeT focuses on observations of each single agent, handling various observation sizes by decoupling the observations into a set of observation-entities according to the physical meanings, matching them with different action-groups, and modeling the relationship between the matched observation-entities with self-attention mechanism for better representation learning [1]. In contrast, MAT focuses more on observations and actions from different agents, modeling the interrelationship between agents with Transformer architectures, promoting the monotonic improvement of joint policies without the requirement for divisible observations. We have introduced this work in Section 2 of the revision and will surely compare with it in the final version (currently completed results are show in this **anonymous** [link](https://anonymous.4open.science/r/MAT_Rebuttal-0ABA/baseline.pdf)).
> >
> > > **Q5** What would happen to MAT's performance if we replaced the transformer by another language model (e.g., GRU, LSTM)?
> >
> > **A5** Theoretically, the multi-agent sequential decision paradigm could be implemented with different type of sequence models including RNNs like GRU and LSTM. However, compared with Transformer, other RNN-based implementations pass the information between sequential tokens with recurrence and will suffer from critical information loss that impedes building long-distance dependencies [2]. In contrast, the attention mechanisms allow the Transformer to model inter-token dependencies regardless of their distance in the input or output sequences.
> >
> > Technically, we conducted an ablation experiment that replaces the encoder and decoder in Figure 2 with GRU, whose results verified the advantage of Transformer architecture (see the **anonymized** [link](https://anonymous.4open.science/r/MAT_Rebuttal-0ABA/structure.pdf) or the *ablation studies* section at the end of our revised appendices).
> > We found this question very insightful and we thank the reviewer for it.
> >
> > **Reference:**
> >
> > [1] Hu, Siyi, et al. "Updet: Universal multi-agent reinforcement learning via policy decoupling with transformers." arXiv preprint arXiv:2101.08001 (2021)
> >
> > [2] Ashish Vaswani, et al. Attention is all you need. In Advances in neural information processing systems, (2017)
> >
> > [3] Yu, Chao, et al. "The surprising effectiveness of ppo in cooperative, multi-agent games." arXiv preprint arXiv:2103.01955 (2021).

---

> > > ### Comment · Reviewer_YpeW · 2022-08-05
> > > **Response to Rebuttal**
> > >
> > > I am satisfied with the rebuttal provided by the authors and the updates they made to the manuscript. I think the proposed approach has demonstrated novelty and its performance is strong compared to other recent methods (QMIX, HAPPO and MAPPO).
> > >
> > > I am raising my score by 1 point.

---

### Public Comment · ~Michael_A._Alcorn1 · 2022-11-21
**baller2vec++ as Relevant Prior Work**

I'm surprised the authors chose not to cite [`baller2vec++`](https://arxiv.org/abs/2104.11980) as relevant prior work, despite this work being brought to their attention on [their reddit post](https://www.reddit.com/r/MachineLearning/comments/v2af3k/comment/ias871o/) in June. `baller2vec++` exploits a chain rule decomposition of the joint *distribution* (instead of policy) of simultaneous agent behaviors to better model multi-agent systems, and similarly uses an autoregressive Transformer over the agents to accomplish this task.

---

> ### Public Comment · Authors · 2022-11-21
> **Response**
>
> Hi Michael:
>
> The essence of our MAT lies in the advantage decomposition theorem and its application in multi-agent RL. Though we agree that baller2vec++ has a similar idea, at a very high level, of conducting sequential style of decision making as our MAT, but they are still very different work. As far as I can tell, baller2vec++ is a way to conquer basketball game, it is not even a RL algorithm, let alone multi-agent reinforcement learning algorithm, therefore, we don't see the necessity of citing this particular work. From MAT's perspective, it is really not relevant to cite baller2vec++ as a general cooperative MARL solution.
>
> Plus, we believe baller2vec++ is not the first work to propose sequential update either, for example, this sequential rollout idea has been proposed by Prof. Dimitri Bertekas [https://arxiv.org/abs/1910.00120] early in 2019, and similar ideas have been discovered again by this work [https://arxiv.org/pdf/2206.07505.pdf] and this work [https://arxiv.org/pdf/2207.11143.pdf].
>
> Last, there are tons of papers published on arXiv. We believe we have no responsibility of going through each one of them and check if they overlaps with our work. We would like to say thank you for bringing the baller2vec++ work to our attention, but we still want to prioritise  citing papers that have been peer-reviewed and published. We will be more than happy to cite your work once it is officially published.

---

> > ### Public Comment · ~Michael_A._Alcorn1 · 2022-11-22
> > **baller2vec++'s Relevance**
> >
> > >From MAT's perspective, it is really not relevant to cite baller2vec++ as a general cooperative MARL solution.
> >
> > The reason `baller2vec++` is relevant is because the paper claims that MAT is a "novel architecture", going as far as to state (emphasis the authors'):
> >
> > >**By treating a team of agents as a sequence**, the Transformer architecture allows us to model teams of agents with variable numbers and types, while avoiding drawbacks of MAPPO/HAPPO.
> >
> > MAT accomplishes this by applying a Transformer with a causal attention mask to a rolled out sequence of agent actions, which is identical to what `baller2vec++` does. The fact that MAT is in a reinforcement learning setting while `baller2vec++` is not is a trivial difference from an architecture standpoint. Even Figure 1 seems to be quite similar to Figure 1 from `baller2vec++` in terms of motivating the architecture.
> >
> > >Last, there are tons of papers published on arXiv. We believe we have no responsibility of going through each one of them and check if they overlaps with our work.
> >
> > I brought `baller2vec++` to your attention in June in the previously mentioned reddit post (in which you said you would cite `baller2vec++`), i.e., it wasn't necessary to go through all of arXiv to discover it.
> >
> > >We would like to say thank you for bringing the baller2vec++ work to our attention, but we still want to prioritise citing papers that have been peer-reviewed and published. We will be more than happy to cite your work once it is officially published.
> >
> > I suppose that's your prerogative. For what it's worth, `baller2vec++` was cited in [a paper by authors from DeepMind](https://www.nature.com/articles/s41598-022-12547-0).
> >
> > I have said all I have to say on this subject, so you're welcome to have the final word.

---

### Public Comment · ~Wenshuai_Zhao1 · 2023-06-02
**Question for the monotonic improvement without waiting preceedor agents' updates**

Hi,

Thanks for the great work! But I am a bit confused by the claim in the paper that
> An MAT agent $i_m$ optimizes a trust-region objective
> that is conditioned on new decisions of agents $i_{1:m-1}$ by means of conditioning its policy ratio on
> them (see Equation (5)). As such, it increases the joint return monotonically like if it followed the
> sequential update scheme of HAPPO.

Does MAT use the actions $\mathbf{a}^{1:m-1}$ from the buffer and perform multiple epoch updates? If so, the actions are already out-of-date and should be reweighted by importance sampling as shown in HAPPO. I wonder how conditioning the $a^m$ to the actions $\mathbf{a}^{1:m-1}$ from the replay buffer without knowing the newest $\bar\pi(\mathbf{a}^{1:m-1}\vert s)$ can satisfy the monotonic guarantee?

Can you explain it further? Thanks in advance and look forward to your reply.

---

> ### Public Comment · ~Muning_Wen2 · 2023-06-03
> **Response**
>
> Hiya,
> Thanks for your interest in our work! Yes, MAT use the collected $a^{i_{1:m-1}}$ to perform policy updates.
>
> At first, we explore the monotonic improvement of joint policies based on the case shown in Figure 1 in HAPPO [1], i.e. agents’ local improvements in performance can jointly result in a worse outcome (conflict), and thus aim to avoid it.
> In deed, the $a^{i_{1:m-1}}$ in buffer will be out-of-data for $\pi(a^{i_{m}}|s, a^{i_{1:m-1}})$ with potential updated actions $\bar{a}^{i_{1:m-1}}$ from $\bar{\pi}(a^{i_{1:m-1}}|s)$, and what we need to update for $i_m$ should be $\pi(a^{i_{m}}|s, \bar{a}^{i_{1:m-1}})$. From the perspective of update with a single sample, it does loosen the monotonic guarantee. However, $\pi(a^{i_{m}}|s, \bar{a}^{i_{1:m-1}})$ and $\pi(a^{i_{m}}|s, a^{i_{1:m-1}})$ are actually independent and could be updated with difference samples (in the same batch, in the past, or in the future). That is, although $a^{i_{1:m-1}}$ is out-of-data, the update for $\pi(a^{i_{m}}|s, \bar{a}^{i_{1:m-1}})$ can happen in other places. Therefore, in terms of final results, MAT can still avoid the conflict and lead to joint actions $\bar{a}^{i_{1:m}}$ better than $a^{i_{1:m}}$ at the end like HAPPO.
>
> As you suggested, reweighing with importance sampling as shown in HAPPO should be a good idea to optimize MAT to rigorously satisfy the monotonic guarantee. But the price to pay is the longer training time so it needs a trade-off according to practical scenarios.
>
> Thank you very much for your constructive questions, which have inspired me to rethink the monotonic guarantees and further optimize this work.
>
> [1] Kuba, J. G., Chen, R., Wen, M., Wen, Y., Sun, F., Wang, J., & Yang, Y. (2021). Trust region policy optimization in multi-agent reinforcement learning. arXiv preprint arXiv:2109.11251.

---

### Meta-Review · Area_Chair_RBYk · 2022-08-29

**Recommendation:** Accept
**Confidence:** Certain

**Metareview:**

The reviewers are largely in consensus that this is a well-executed application of a highly relevant technology (transformers) to an interesting and challenging problem, with pleasing results. I believe the community will be quite interested to see this development presented at NeurIPS.

Note: I agree with reviewer MEiB that important prior work also using transformers needs careful comparison, but the author's rebuttals gave confidence that will be done appropriately. Please do be very dilligent on this point in the final version.



**Award:**

No

---

### Decision · Program_Chairs · 2022-09-14

Accept